# PARP1 promotes gene expression at the post-transcriptional level by modulating the RNA-binding protein HuR

Yueshuang Ke[1,2], Yanlong Han[2], Xiaolan Guo[2], Jitao Wen[2], Ke Wang[2], Xue Jiang[2], Xue Tian[2], Xueqing Ba[1,2], Istvan Boldogh[3] & Xianlu Zeng[1,2]

Poly(ADP-ribosyl)ation (PARylation) is mainly catalysed by poly-ADP-ribose polymerase 1 (PARP1), whose role in gene transcription modulation has been well established. Here we show that, in response to LPS exposure, PARP1 interacts with the adenylateuridylate-rich element-binding protein embryonic lethal abnormal vision-like 1 (Elavl1)/human antigen R (HuR), resulting in its PARylation, primarily at site D226. PARP inhibition and the D226 mutation impair HuR's PARylation, nucleocytoplasmic shuttling and mRNA binding. Increases in mRNA level or stability of pro-inflammatory cytokines/chemokines are abolished by PARP1 ablation or inhibition, or blocked in D226A HuR-expressing cells. The present study demonstrates a mechanism to regulate gene expression at the post-transcriptional level, and suggests that blocking the interaction of PARP1 with HuR could be a strategy to treat inflammation-related diseases that involve increased mRNA stability.

[1] The Key Laboratory of Molecular Epigenetics of the Ministry of Education, Northeast Normal University, Changchun, Jilin 130024, China. [2] Institute of Genetics and Cytology, Northeast Normal University, Changchun, Jilin 130024, China. [3] Department of Microbiology and Immunology, Sealy Center for Molecular Medicine, University of Texas Medical Branch at Galveston, Galveston, Texas 77555, USA. Correspondence and requests for materials should be addressed to X.B. (email: baxq755@nenu.edu.cn) or to X.Z. (email: zengx779@nenu.edu.cn).

Poly(ADP-ribosyl)ation (PARylation) is an essential post-translational protein modification catalysed by poly-ADP-ribose polymerases (PARPs), a family of enzymes that polymerize ADP-ribose units from NAD$^+$ and transfer the polymer known as poly-ADP-ribose (PAR) onto a variety of proteins[1]. PARP1 is currently the best understood member of the PARP family, and is affirmed as accounting for at least 85% of cellular PARP activity[2]. PARP1 has been implicated in a wide range of biological processes, such as maintenance of genome integrity, transcriptional regulation, energy metabolism and cell death[3,4]. Although originally characterized as a key factor in DNA repair and cell death pathways, PARP1's role in regulation of gene expression under basal and signal-activated conditions has been demonstrated by a wealth of studies[5,6]. Extensive studies have documented that the transcriptional activation constitutes the primary mode of PARP1 modulating gene expression. PARylation, which introduces massive negative charges to the linker histone H1 and core histones[1,3,7,8], mediates the relaxation of the chromatin superstructure and then facilitates the recruitment of transcription machinery to the promoters or enhancers of target genes. In addition, PARP1 is involved in the activation of transcription factors such as nuclear factor-kappa B (NF-κB), activator protein 1 (AP-1) and heat-shock factor protein 1 to regulate gene expression[9]. A large number of studies have well addressed the involvement of PARP1 activation in inflammatory disorders via PARP1-dependent upregulation of pro-inflammatory genes[9]. Our previous studies reported that PARP1 binds to and modifies RelA/p65 (refs 9–11) and, therefore, promotes the NF-κB-dependent expression of pro-inflammatory cytokines.

The expression of inflammatory genes is tightly regulated by both transcriptional and post-transcriptional mechanisms because modifying messenger RNA (mRNA) stability provides rapid and flexible control, and is particularly important in coordinating the initiation and resolution of inflammation[12]. This urged us to investigate whether PARP1 regulates the expression of inflammatory cytokines/chemokines at the post-transcriptional level. Emerging data have revealed the roles of PARP1 in RNA metabolism. An intriguing study showed that poly(A) polymerase is PARylated during heat shock, leading to the inhibition of mRNA polyadenylation of target genes in a PARP1-dependent manner[13]. In the present study, macrophages were exposed to lipopolysaccharide (LPS) with or without PARP1 inhibition. Our results showed LPS-induced increase in the stability of mRNAs from pro-inflammatory genes including Cxcl2 is diminished by PARP1 inhibition/depletion. PARP1 interacts with the adenylateuridylate-rich element (ARE)-binding protein embryonic lethal abnormal vision-like 1 (Elavl1)/human antigen R (HuR) resulting in its PARylation. The increased PARylation of HuR enhances nucleocytoplasmic shuttling and mRNA binding, and promotes mRNA stability. The results presented a mechanism to regulate gene expression at the post-transcriptional level by PARP1 activation.

## Results

**PARP1 augments Cxcl2 expression at post-transcriptional level.** To determine the stability of mRNA, a classical approach[14] was used as illustrated in Supplementary Fig. 1a. Briefly, parallel cultures of murine primary peritoneal macrophages (pMφ) were exposed to 500 ng ml$^{-1}$ LPS for 1 h to boost pro-inflammatory gene expression, and then the transcription inhibitor actinomycin D (Act D) was added in media with or without LPS (±PARP inhibitor PJ34) for 4 h. The levels of remaining mRNAs were determined using Mouse Inflammatory Cytokines & Receptors PCR arrays (SABiosciences). In response to LPS, the mRNA

stability of the most tested inflammatory mediators was increased, especially those encoding chemokine receptors (for example, Ccrs), C-C (for example, Ccl11) and C-X-C (for example, Cxcl1 and Cxcl13) chemokines, as well as interleukins (for example, IL1β) (Supplementary Fig. 1b,d). LPS-induced increases in the remaining mRNA levels were significantly abolished by PJ34. For example, levels of Cxcl1, Ccl11, Cxcl13 and Il1β were decreased by 2.14-, 2.17-, 3.16- and 2.29-fold, respectively (Supplementary Fig. 1c,e). Interestingly, the levels of some Ccrs (for example, Ccr4, 5, 6, 7 and 8), Cxcrs (for example, Cxcr2 and 5) and cytokines/chemokines (for example, Ifn and Cxcl11) were not affected by PARP's inactivation (Supplementary Fig. 1c,e). Cxcl1 and Cxcl2 (homologues of human growth-regulated protein (Gro) α and β, respectively) are potent attractants of neutrophils, highly relevant to innate inflammatory responses[15,16], thus real-time PCR was performed to examine their remaining mRNAs individually. LPS stimulation induced ∼2.5- and ∼4.5-fold increases in the levels of remaining Cxcl1 and Cxcl2's mRNA, respectively, which were diminished by PJ34 administration to the basal levels (Supplementary Fig. 1f). These results verified the involvement of PARP1 in mRNA stability regulation, and also suggested Cxcl2 mRNA more susceptible to be affected by PARP1 activation.

Next, we examined the kinetics of the level of remaining Cxcl2 mRNA. The half-life of Cxcl2 mRNA in LPS-withdrawn cells significantly declined after Act D addition, whereas it was sustained in LPS-stimulated cells. PJ34 administration abrogated the increase in Cxcl2 mRNA stability induced by LPS. A two-way analysis of variance analysis indicated a significance of $P < 0.001$. (Fig. 1a,b). Other PARP1 inhibitors, 3-aminobenzamide (3-AB) and Olaparib, exhibited the same effect (Supplementary Fig. 2a). Inhibitor targeting-off effects (for example, unspecific block of TLR/inflammasome signalling) were excluded as PJ34 did not impair the LPS-induced IRAK1 phosphorylation (Supplementary Fig. 2b). Moreover, when PARP1 expression was silenced, the remaining Cxcl2 mRNA in LPS-stimulated cells was decreased to 40%, compared with that of control short interfering RNA (siRNA)-transfected cells (Fig. 1c). Small interfering RNA targeting another sequence of PARP1 showed a similar result (Supplementary Fig. 2c). Knockdown of PARP2, a functional back-up of PARP1, had no impact on the level of remaining Cxcl2 mRNA (Supplementary Fig. 2d), specifying the role of PARP1 in maintaining mRNA stability.

The AREs commonly existing in the 3′-untranslated regions (UTRs) are major mRNA destabilization determinants[17,18]. With their binding proteins, AREs have significant physiological functions in the modulation of mRNA stability. Cxcl2 mRNA contain tandem overlapping repeats of AUUUA motifs (class I AREs)[19]; therefore, we investigated the implication of 3′-UTR in the modulation of Cxcl2 mRNA stability by PARP1. A reporter plasmid was constructed as described previously[20] (Fig. 1d). Dual-reporter assays revealed firefly luciferase activity in cells transfected with a Cxcl2-3′-UTR construct was severely impaired (to ∼3.8%) compared with that in cells transfected with pGL3-control (Fig. 1e). In parallel experiments, the firefly luciferase mRNA level was significantly decreased (to ∼3%) (Fig. 1f), indicating the Cxcl2-3′-UTR was indeed a destabilizing determinant[21]. LPS stimulation did not affect the activity or the mRNA level of firefly luciferase in cells transfected with the pGL3-control, where the luciferase gene was constitutively transcribed. Importantly, LPS stimulation significantly increased firefly luciferase activity (∼3.5-fold), as well as its mRNA level (∼5-fold) in cells transfected with the Cxcl2-3′-UTR construct, which, however, were markedly diminished by PJ34 (Fig. 1e,f). The combined data implied

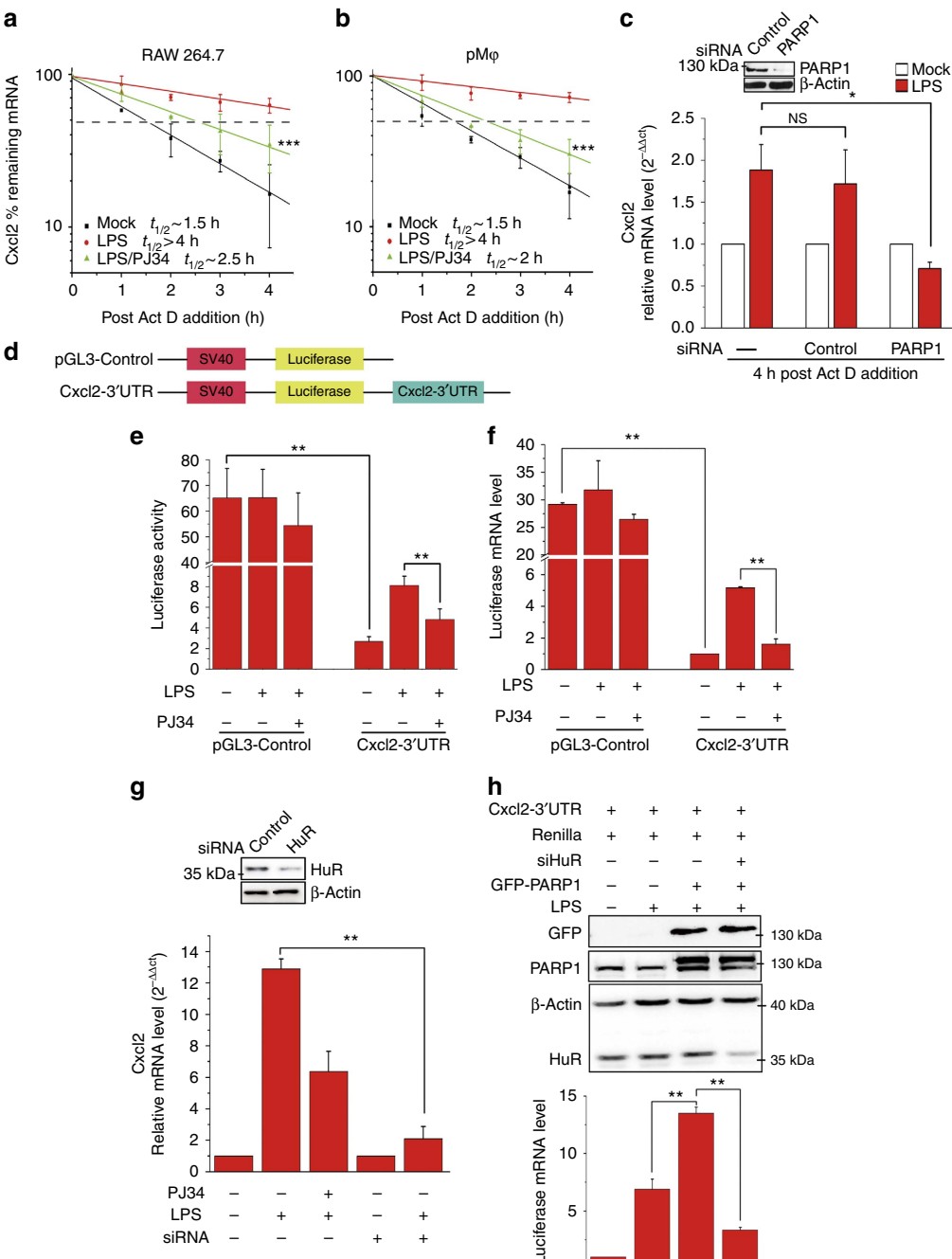

**Figure 1 | PARP1 augments Cxcl2 expression at the post-transcriptional level via HuR.** (**a,b**) PARP1 activity is essential for the increase in *Cxcl2* mRNA half-lives in LPS-exposed macrophages. RAW 264.7 and pMφ cells were exposed to LPS for 1 h and then subjected to transcriptional inhibition with or without LPS maintenance (± PJ34) for various lengths of time as indicated. Real-time PCR was performed to assess the remaining *Cxcl2* mRNA levels. The half-lives of different samples are indicated in the inset. A two-way analysis of variance (ANOVA) indicated the significance between the LPS/PJ34 and LPS groups at ***$P < 0.001$. (**c**) The depletion of PARP1 decreases *Cxcl2* mRNA stability. RAW 264.7 cells were transfected with PARP1 or control siRNA, and 48 h later, they were exposed to LPS and subjected to transcriptional inhibition with or without LPS maintenance (± PJ34) for 4 h. Real-time PCR was used to assess the remaining *Cxcl2* mRNA levels. (**d**) Diagram of the *Cxcl2*-3′-UTR reporter plasmid construct. (**e,f**) PARP1 regulates *Cxcl2* mRNA stability through its 3′-UTR. RAW 264.7 cells were transfected with reporter plasmids containing the *Cxcl2*-3′-UTR or vector pGL3-control plus the Renilla reporter plasmid, and then challenged with LPS for 5 h. Luciferase activity (**e**) and its mRNA levels were analysed by real-time PCR (**f**). (**g**) HuR is involved in the increase in *Cxcl2* mRNA level induced by LPS stimulation. RAW 264.7 cells were transfected with siRNA targeting HuR or the control, and then challenged with LPS for 5 h. *Cxcl2* mRNA levels were determined by real-time PCR. (**h**) HuR mediates the role of PARP1 in mRNA stabilization. HEK 293 cells were subjected to HuR silencing, or not, and then co-transfected with the above-mentioned *Cxcl2*-3′-UTR dual-reporter system with or without the GFP-PARP1 plasmid. Cells were cultured normally and challenged with or without LPS. The levels of luciferase mRNA were determined by real-time PCR using Renilla luciferase mRNA for calibration. Data were expressed as mean ± s.d. ($n = 5$), and analysed by one-way ANOVA. *$P < 0.05$, **$P < 0.01$, NS, not significant.

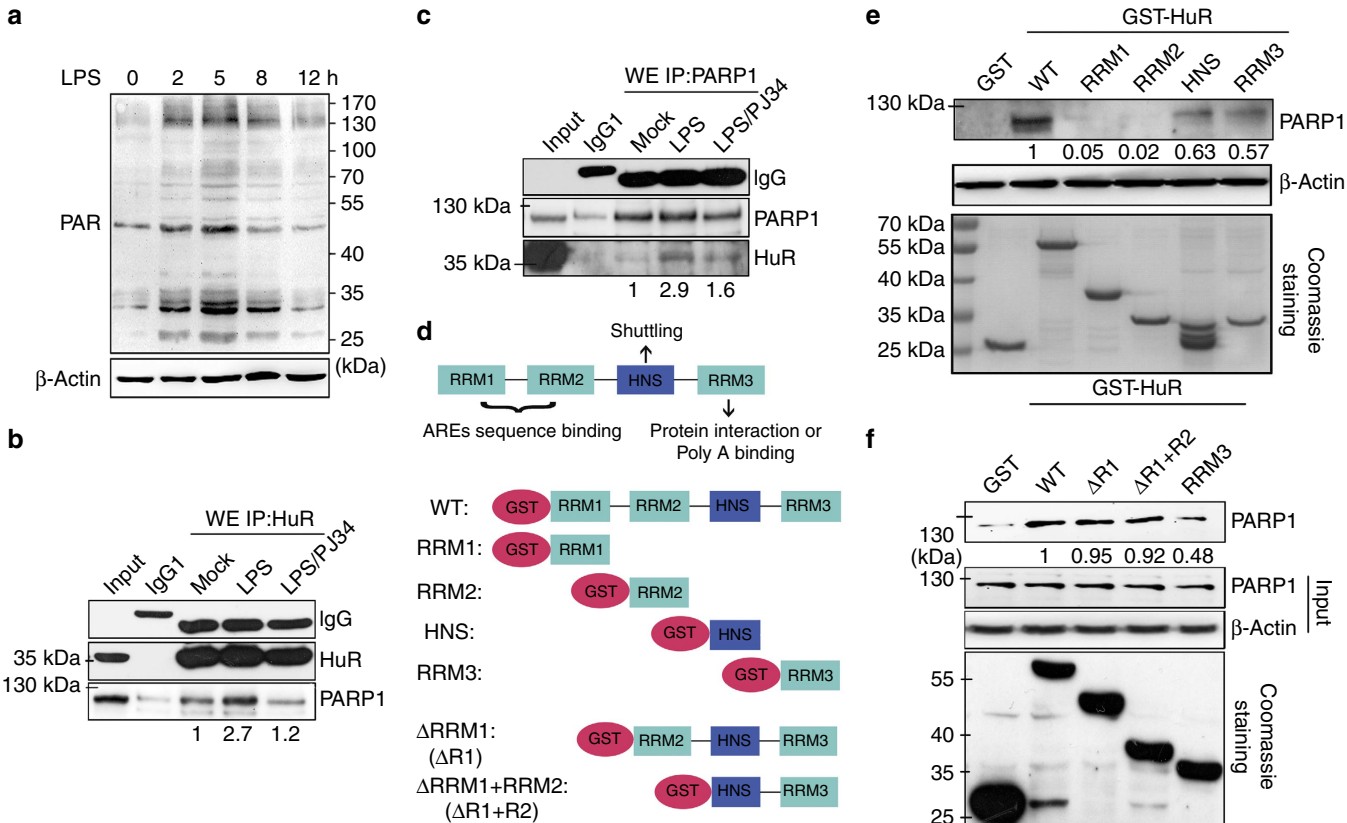

**Figure 2 | PARP1 interacts with HuR in response to LPS exposure.** (**a**) LPS stimulation promotes protein PARylation. RAW 264.7 cells were challenged with LPS for various lengths of time. Immunoblotting was performed to detect the PARylation levels of proteins in whole-cell lysates. (**b,c**) LPS exposure increases the association of PARP1 with HuR. RAW 264.7 cells were mock-treated or LPS-exposed (± PJ34) for 5 h. Whole-cell extracts (WEs) were prepared and immuno-precipitates were obtained using antibodies recognizing HuR (**b**) and PARP1 (**c**). The association of PARP1 with HuR was detected by immunoblotting. (**d**) Diagram of the domains in HuR. The schematics of the GST-HuR expression plasmid, as well as domains and truncated mutants. (**e,f**) HNS and RRM3 mediate the association of HuR with PARP1. GST and GST-HuR, as well as the domain (**e**) and truncated (**f**) mutants were incubated with equal amounts of WEs from LPS-treated cells. Levels of pulled down PARP1 were detected by immunoblotting.

that *Cxcl2*'s 3′-UTR mediates PARP1's regulation of mRNA stability.

Among ARE-binding proteins, HuR, a ubiquitously expressed member of the ELAVL family of proteins, is one of the few that have been demonstrated to stabilize ARE-containing mRNAs[22–24]. In our present study, the silencing of HuR abrogated LPS-induced increases in *Cxcl2* mRNA levels (Fig. 1g; Supplementary Fig. 2e). To verify that HuR mediates the role of PARP1 in stabilizing *Cxcl2* mRNA, siRNA-targeting HuR or controls were delivered into cells, and then a co-transfection of the GFP-PARP1 plasmid with the above-mentioned *Cxcl2*-3′-UTR dual-reporter system was conducted. Cells were either exposed to LPS or not, and firefly mRNA levels were determined by real-time PCR using Renilla luciferase mRNA for calibration. LPS-induced increase in firefly mRNA level was enhanced by the overexpression of PARP1, which was eliminated by HuR silencing (Fig. 1h). Tristetraprolin, one of the dominant mRNA-destabilizing factors, was not shown to be distributed in the nuclei with or without LPS stimulation in the present study, excluding the possibility of it mediating PARP1's role (Supplementary Fig. 2f). The combined data suggested that PARP1 modulates *Cxcl2* mRNA stability by acting on HuR.

**HuR is associated with PARP1 in response to LPS stimulation**. To gain insights into the molecular mechanism how PARP1 modulates mRNA stability via HuR, we first examined the dynamics of protein PARylation and the interaction of PARP1 with HuR in cells exposed to LPS. The content of PARylated

proteins notably increased at 2 h and reached a maximum level at 5 h post LPS addition (Fig. 2a). Thus, cells were collected after 5 h of LPS stimulation, and co-immunoprecipitation (Co-IP) assays were performed. Results showed a low-level interaction of HuR and PARP1 in the extract of untreated cells. The interaction of the two proteins was increased upon LPS stimulation, which was significantly inhibited by PJ34 (Fig. 2b,c). Both PARP1 and HuR are targets of caspases and may undergo cleavage under stress. Thus, the whole-cell lysate was applied along with a molecular weight standard, which indicated that the interaction between the two molecules requires their full-length forms (Fig. 2b,c). In addition, in the presence of RNase, the HuR–PARP1 complex did not collapse, suggesting that the association of PARP1 with HuR is not mediated by HuR-bound RNA (Supplementary Fig. 3a).

Next, we asked what domain(s) of HuR interact(s) with PARP1. HuR has three RNA recognition motifs (RRMs) through which it interacts with target mRNAs and partner proteins[25]. Located between RRM2 and RRM3 is a hinge region that encompasses a nucleocytoplasmic shuttling sequence (HNS), spanning residues 205–237 (ref. 26; Fig. 2d, upper). GST-HuR prokaryotic expression plasmid and domain mutants were constructed (Fig. 2d, middle), and pull-down assays were performed. While GST could not pull down any PARP1, GST-HuR exhibited a significant interaction. GST-HuR-HNS and GST-HuR-RRM3 could modestly pull down PARP1, whereas GST-HuR-RRM1 and GST-HuR-RRM2 barely showed such

an interaction (Fig. 2e). Furthermore, an N-terminal truncated mutation was made (Fig. 2d, lower), and the pull-down assay showed that the deletion of RRM1 and RRM2 did not exhibit an obvious impact on the interaction of HuR with PARP1, whereas the absence of the HNS did (Fig. 2f). The combined results suggested that the interaction of HuR and PARP1 depend on the HNS and RRM3 domains.

**PARP1 PARylates HuR primarily at the aspartic acid 226.** We probed the HuR-associated complex precipitated from cell lysates with an antibody against PAR and probed the PAR-associated proteins with an antibody against HuR. As we expected, HuR was PARylated in extracts of LPS-exposed cells (Fig. 3a). Full membranes with molecular weight standards are shown in Supplementary Fig. 3b. Along with the absence of severe DNA damage, the length of the PAR polymer is considerably shorter, ranging from a single residue to oligo units[6]. PARylated HuR did not exhibit apparent shift retardation, which was also noticed with other PARylated mRNA metabolism-related proteins[27]. In addition, we prepared cytosolic extract (CE) and nuclear extract (NE) to perform IP assays. Results showed that the interaction of HuR with PARP1 only occurred in NE fractions and displayed a similar pattern to that shown from whole-cell lysates (Fig. 3b, right). Intriguingly, PARylation patterns of HuR in CE and NE fractions exhibited notable differences. In CE fractions, LPS stimulation resulted in an increase in PARylated HuR, which was markedly decreased by PJ34 (Fig. 3b, left). Whereas the levels of PARylated HuR in NE fractions exhibited a moderate increase compared with that in CE, which might be a consequence of nucleocytoplasmic shuttling of PARylated HuR. PJ34 also significantly inhibited HuR's PARylation in NE fractions (Fig. 3b, right). The combined results suggested that LPS stimulation enhanced the interaction of HuR and PARP1, which led to an increase in PARylation of HuR.

To further address which domain(s) and site(s) are PARylated, we developed an *in vitro* PARylation assay using GST-fused proteins as described in the Methods. First, bead-coated GST and GST-HuR were incubated with or without PARP1 in the presence or absence of PJ34 or Poly(ADP-ribose) glycohydrolase (PARG), the enzyme removing ADP-ribose units from the target proteins[28,29] (Fig. 3c, left). Incubation with PARP1 resulted in strong modifications of GST-HuR (compare lanes 2 and 3), but not GST (lane 1). In the presence of PJ34 (lane 4) or PARG (lane 5), the modifications of HuR were diminished. If the soluble GST and GST-HuR eluted from the beads were applied (Fig. 3c, right), the PARylation patterns of GST-HuR were similar to that occurred on bead-bound GST-HuR (lanes 7–10). However, a notable difference was observed when GST was incubated with PARP1 (compare lanes 1 and 6). While no signal was detectable below 100 kDa, a strong smear was exhibited at the top of the lane (which also could be observed in lanes 3 and 8), indicating the autoPARylated PARP1. The combined results verified the activation of PARP1 and the specificity of HuR PARylation. Next, domain mutants were studied, and GST-HuR-HNS was strongly PARylated (Fig. 3d). Then, we further questioned which potential amino acid, lysine 191 (K191) or aspartic acid 226 (D226), was modified. An alanine substitution mutation was created (Fig. 3e). PARylation assays showed the D226, but not the K191 mutation blocked the PARylation of GST-HuR-HNS (Fig. 3f). The D226A mutation also caused barely detectable PARylation of the full-length GST-HuR (Fig. 3g). The combined results suggested that D226 is the primary site of PARylation.

**PARP1 promotes the LPS-induced shuttling of HuR.** It has been documented that HuR's function is regulated primarily at the level of nucleocytoplasmic shuttling[19]; hence we examined the effect of PARylation on HuR's distribution. Immuno-fluorescence (IF) staining showed that HuR was localized in nuclear compartments in mock-treated cells, while in LPS-exposed RAW 264.7 (Fig. 4a) and pMφ cells (Supplementary Fig. 4a), apparently distributed to the cytoplasm. To quantify the nucleocytoplasmic shuttling of HuR, immunoblotting was performed (Supplementary Fig. 4b). Both approaches showed that the cytoplasmic localization of HuR peaked at 5 h and lasted up to 8 h after LPS stimulation, coinciding with protein PARylation kinetics (Fig. 2a). LPS-induced HuR's nuclear export was blocked by PJ34 (Fig. 4b; Supplementary Fig. 4c,d). With the increase in cytoplasmic HuR, no effective reduction of nuclear HuR occurred (Supplementary Fig. 4b), implying the increased expression of HuR in response to LPS, which was verified by a time kinetics analysis of HuR expression (peaked at 5 h post LPS addition; Supplementary Fig. 4e). Thus, to exclude that an increase in the cytoplasmic HuR level is the consequence of enhanced protein synthesis, protein synthesis inhibitor cycloheximide (CHX, $10 \mu g \, ml^{-1}$) was applied after 1 h LPS exposure. IF staining of HuR in CHX-applied cells supported the role of PARP1 in promoting HuR's nuclear export (Fig. 4c; Supplementary Fig. 4f).

To specify the implications of PARP1, PARP1 silencing was carried out. In cells transfected with control siRNA, LPS stimulation induced a marked nuclear export of HuR (Fig. 4d, left). Whereas, when cells were transfected with siRNA-targeting PARP1, LPS-induced nucleocytoplasmic shuttling of HuR occurred only in the cells where PARP1 failed to be effectively depleted (Fig. 4d, right, note yellow arrows). When PARP1 was successfully silenced, the shuttling of HuR was nearly thoroughly blocked (Fig. 4d, right, note white arrows). These results suggested PARP1's indispensable role in regulating the shuttling of HuR. An immunoblotting analysis provided the quantification of HuR's cytoplasmic redistribution due to PARP1 interference (Supplementary Fig. 4g).

**Protein PARylation enhances binding of HuR to *Cxcl2* mRNA.** Binding of HuR's to mRNA counteracts the destabilizing effects of tristetraprolin, TFIIB-related factor 1, KH-type splicing regulatory protein and AU-binding factor 1 (ref. 19), accounting for another aspect of its roles in mRNA protection. Thus, we performed RNA-immunoprecipitation (RNA-IP) assays as described in Methods. From the whole-cell lysate of mock-treated cells, the binding of HuR to *Cxcl2* mRNA was barely detectable; however, from the LPS-challenged cell lysate, abundant *Cxcl2* mRNA was pulled down. The interaction of HuR with *Cxcl2* mRNA was inhibited by PJ34 (Fig. 5a). Given that HuR undergoes nucleocytoplasmic shuttling upon activation, we prepared CEs for RNA-IP assays. In the CE fraction from LPS-stimulated cells, the HuR-associated *Cxcl2* mRNA level was higher than that from mock-treated cells, which was decreased by PJ34 (Fig. 5b), the levels of a set of ARE-containing mRNAs, such as *Cxcl1*, *Cxcl13* and *Il-1β*, showed the similar patterns in HuR's immuno-precipitates (Supplementary Fig. 5). Intriguingly, as a control, *Ccr7* mRNA, whose stability was not subjected to PARP1 regulation as shown by Inflammatory Cytokines & Receptors PCR arrays (Supplementary Fig. 1c,e), could not be pulled down with HuR (Fig. 5c). In support, the remaining level of *Ccr7* mRNA was increased upon LPS exposure but not affected by PJ34 (Fig. 5d). The data suggested that protein PARylation increases the association of HuR with the mRNA targets.

**PARP1 enhances the binding of HuR to ARE-containing RNA.** To further confirm that the binding of HuR to the ARE motif is regulated by PARylation, GST-HuR was used to perform

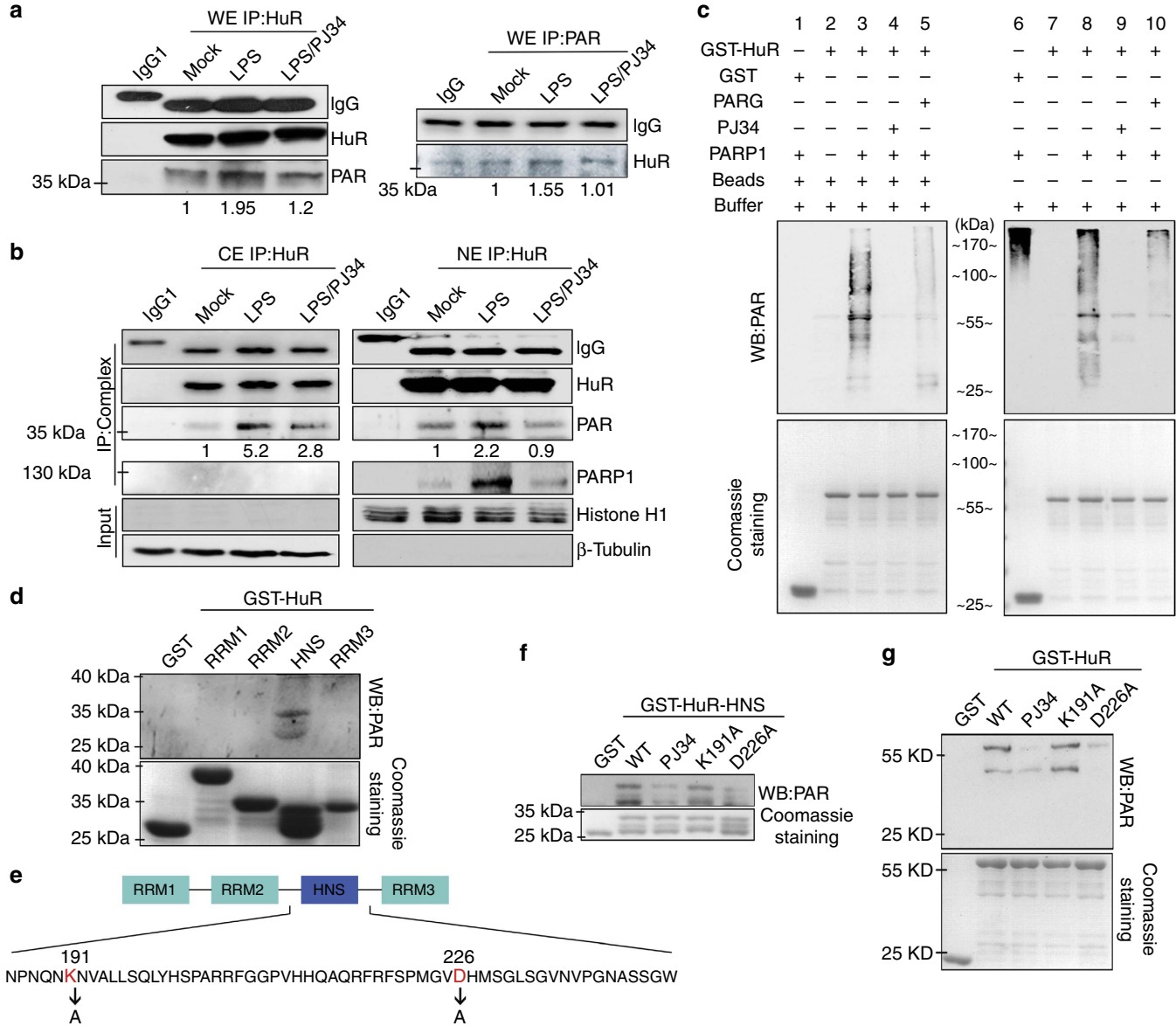

**Figure 3 | PARP1 PARylates HuR primarily at aspartic acid 226 upon LPS stimulation. (a)** LPS exposure increases the PARylation level of HuR. RAW 264.7 cells were mock-treated or LPS-exposed (± PJ34) for 5 h. Immuno-precipitates were prepared using antibodies recognizing HuR (left) or PAR (right). The PARylation of HuR was detected using an antibody against PAR (left) or HuR (right). (**b**) LPS induces the PARylation of HuR in nuclear and cytoplasmic compartments. RAW 264.7 cells were treated differently as described in the legend to **a**. Cytosolic (CEs) and nuclear (NEs) extracts were prepared, and IP assays were performed using the HuR antibody. HuR's binding with PARP1, and its PARylation, was assessed by immunoblotting. (**c**) HuR can be PARylated by PARP1 *in vitro*. Equal amounts of bead-coated (left) or soluble (right) GST and GST-HuR were incubated with or without PARP1 in the presence or absence of PJ34 or PARG, and then subjected to immunoblotting to detect PARylation levels using Ab against PAR. (**d**) HNS is the major domain of HuR that is PARylated by PARP1. Equal amounts of GST-HuR domain mutants were incubated with recombinant protein PARP1 and then subjected to immunoblotting to detect PARylation levels. (**e**) The potential sites subjected to PARylation in the HNS domain of HuR. (**f,g**) D226 is the PARylation site of HuR. Equal amounts of wild-type (WT) GST-HuR-HNS, as well as K191 and D226 mutants (**f**), and full-length GST-HuR, as well as K191 and D226 mutants (**g**), were incubated with recombinant PARP1 protein and then subjected to immunoblotting to detect PARylation levels. Lower panels in **c,d,f** and **g** show the substrate amounts of recombinant GST and GST-fused proteins stained with Coomassie brilliant blue.

RNA electrophoretic mobility shift assays (EMSA). GST-HuR or GST was purified and eluted, and then subjected to PARylation or not, followed by incubation with biotin-labelled tandem ARE repeat-containing RNA oligos. GST-HuR elicited several shifted bands, which might result from the various copies of GST-HuR harboured on the tandem ARE-containing probes (Fig. 6a, lanes 1–12), whereas GST failed to do so (Fig. 6a, lane 13). The incubation with PARP1 markedly enhanced the binding of GST-HuR with the probes (Fig. 6a, compare lanes 4–6 with lanes 1–3), which was inhibited by PJ34 (Fig. 6a, lanes 7–9).

The direct incubation of eluted GST-HuR with probes showed the same patterns of the shifted bands (Fig. 6a, lanes 10–12) as that of samples subjected to PARylation (Fig. 6a, lanes 1–9); in parallel, as a vehicle control, PARP1 in PARylation buffer alone did not result in any shifted bands (Fig. 6a, lane 14). The addition of titrated cold probe resulted in a dose-dependent competition (Fig. 6b), indicating the specificity of the binding of GST-HuR. Furthermore, an antibody supershift assay was performed (Fig. 6c). Both HuR and GST antibodies led to super shifts from the protein–probe complexes (asterisk-labelled) while PARP1

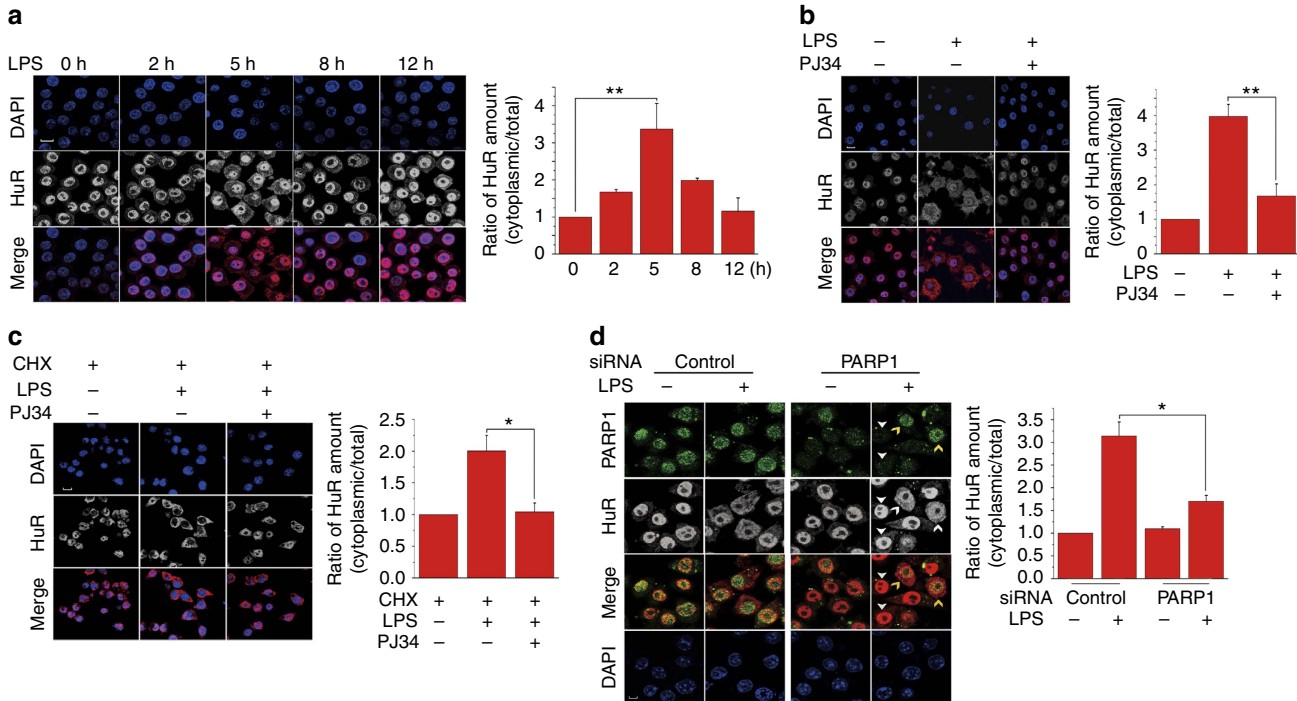

**Figure 4 | Role of PARP1 in HuR's nucleocytoplasmic shuttling in LPS-exposed cells. (a)** LPS stimulation induces the nuclear export of HuR. RAW 264.7 were challenged with LPS for various lengths of time. The subcellular distribution of HuR is determined by immune-fluorescence (IF) staining. Scale bar, 10 μm. **(b)** PARP1 inactivation blocked HuR's nucleocytoplasmic shuttling, which was induced by LPS. RAW 264.7 cells were mock-challenged or LPS-exposed ( ± PJ34) for 5 h. The subcellular distribution of HuR was determined by IF staining. Scale bar, 10 μm. **(c)** The increased cytoplasmic level of HuR is not due to its enhanced protein synthesis. RAW 264.7 cells were exposed to LPS for 1 h, and then subjected to protein synthesis inhibition by the addition of cycloheximide (CHX, 10 μg ml$^{-1}$). Cells withdrawn from LPS (left), maintained in LPS incubation (middle) or treated with LPS + PJ34 (right) were fixed, and IF staining was conducted to detect the distribution of HuR. Scale bar, 10 μm. **(d)** PARP1 silencing prevents HuR's nucleocytoplasmic shuttling that is induced by LPS. RAW 264.7 cells were transfected with control or siRNA-targeting PARP1 and then challenged with LPS for 5 h. The expressions and distributions of PARP1 and HuR were visualized by IF staining. Scale bar, 10 μm. The cytoplasmic distribution of HuR was quantified by densitometry analysis using Image J software (version 1.44; right panels) as described in Methods. *$P < 0.05$, **$P < 0.01$.

antibody failed to do so, similar to IgG1. Recently, PARP1 was reported to directly interact with noncoding pRNA, binding to silent ribosomal RNA genes after their replication in the mid-late S phase[30]. We questioned whether the absence of interaction between PARP1 and RNA in the present study is due to strong binding of PARP1 to the sonicated DNA. We performed a gel-shift assay with titrated sonicated DNA. No shifted bands appeared with the decreasing amount of sonicated DNA (Fig. 6d), verifying no direct binding of PARP1 with the probes. Because the binding of PARP1 with noncoding pRNA relies on a hairpin structure[30], we deduced PARP1 may not able to bind with single-stranded RNA.

Furthermore, isometric RNA oligos containing three AREs that exist in the native UTR domains of *Cxcl2* mRNA were designed, as illustrated in Fig. 7a. An RNA-EMSA showed that GST-HuR could interact with ARE1- and ARE2-, but not ARE3-containing probes (Fig. 7b Lanes 2, 6 and 10), and incubation with PARP1 enhanced the binding of GST-HuR (Fig. 7b lanes 3 and 7), which was abolished by the PJ34 (Fig. 7b, lane 4 and 8). As a control, GST was not able to interact with any of the ARE motifs even after incubation with PARP1 (Fig. 7b, lanes 1, 5 and 9). An antibody supershift assay and cold probe competition verified the specificity of HuR's binding (Fig. 7c). Furthermore, we performed an EMSA to compare the binding of wild-type (WT) and D226A HuR to *Cxcl2*-ARE1 RNA oligo, and the $K_d$ values were calculated as previously described[31]. After incubation with PARP1, WT HuR's binding to the *Cxcl2*-ARE1 RNA oligo increased by more than two folds ($K_d$ value decreased to ~40%), while the D226A mutant displayed similar affinity kinetics to those of

WT HuR without incubation with PARP1 (Fig. 7d). The combined data verified the role of PARP1 in binding of HuR to ARE-containing mRNAs.

**D226 PARylation is crucial for HuR's function.** To gain further insight into the physiopathological significance of D226-mediated HuR PARylation in an intracellular context, we constructed eukaryotic expression plasmids expressing WT Flag-HuR, as well as D226A and K191A mutants. Due to the efficiency of transfection, HEK 293/hTLR4A-MD2-CD14 Cells (HEK 293 cells stably transfected with the human *TLR4*, *MD2* and *CD14* genes) were utilized. Both human and murine *Cxcl2* mRNAs contain AREs in their 3′-UTRs. IP assays using antibody recognizing Flag tag revealed that WT Flag-HuR and the K191A mutant were highly PARylated in LPS-exposed cells, whereas the D226A mutant was not (Fig. 8a). An RNA-IP assay revealed that LPS stimulation increased WT and K191A Flag-HuR-bound *Cxcl2* mRNA levels, but not that with the D226A mutant (Fig. 8b). IF staining and immunoblotting analysis of both RAW 264.7 and HEK 293 cells further affirmed WT and the K191A mutant, but not D226A Flag-HuR, were able to shuttle to the cytoplasm upon the LPS challenge (Fig. 8c,d; Supplementary Fig. 6a,b). Importantly, the ectopic expression of murine WT and D226A HuR in endogenous HuR-silenced HEK 293 cells showed that HuR depletion strongly blocked LPS-induced increases in *CXCL2* mRNA level, which was markedly rescued by the overexpression of murine WT but not D226A HuR (Fig. 9a,b). Further investigations into the stability of the pro-inflammatory gene's mRNA

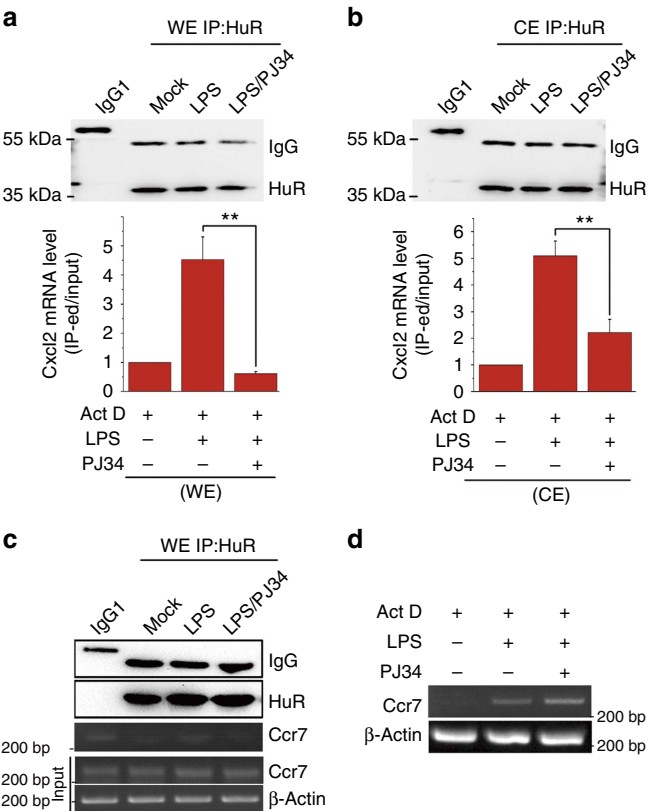

**Figure 5 | Protein PARylation promotes the binding of HuR to *Cxcl2* mRNA.** (**a,b**) PARP1 activation is involved in the LPS-induced binding of HuR to *Cxcl2* mRNA. RAW 264.7 cells were exposed to 500 ng ml$^{-1}$ LPS for 1 h to boost pro-inflammatory gene expression, followed by the addition of Act D. Meanwhile, the cells were withdrawn from LPS, maintained in LPS or treated with LPS + PJ34 for another 2 h. Whole-cell lysates (**a**) and cytolic extracts (CEs) (**b**) were prepared. RNA-IP was conducted using a HuR antibody. Half of the bead–antibody–protein/mRNA complexes were utilized for immunoblotting to assess equal loading/input of HuR, and the remaining half was subjected to real-time PCR to detect pull-down *Cxcl2* mRNA levels using that in the whole-cell lysate for calibration. (**c**) *Ccr7* mRNA is absent in the HuR-associated complex that comes from cells exposed to LPS. RAW 264.7 cells were treated differently, and an RNA-IP was conducted using a HuR antibody as described in the legend to **a** and **b**. HuR-associated *Ccr7* mRNA levels were shown by PCR with reverse transcription (RT)–PCR and electrophoresis. (**d**) *Ccr7* mRNA stability is not related to PARP1 activity. RAW 264.7 cells were exposed to LPS stimulation for 1 h and then subjected to transcriptional inhibition with the maintenance of the LPS challenge (± PJ34), or not, for 4 h. RT–PCR and electrophoresis were conducted to detect *Ccr7* mRNA levels. Data were expressed as mean ± s.d. (n = 5), and analysed by one-way analysis of variance. **P < 0.01.

showed that the half-life of *CXCL2* mRNA in WT HuR-expressing cells was ∼4 h, which was reduced to ∼2 h in D226A HuR-expressing cells (Fig. 9c,d). Also, the stability of *CXCL1*, *IL-8* and *TNFα* mRNAs was significantly lower in D226A mutant-expressing cells than in WT HuR-expressing ones (Supplementary Fig. 7). The results indicated the functional significance of D226 PARylation of HuR in response to immune stimulation.

**LPS induces HuR interaction with PARP1 and is PARylated *in vivo*.** To investigate the inducible interaction of PARP1 with, and PARylation of HuR in an *in vivo* scenario, mice lungs were

exposed to LPS through an intranasal route with or without a PJ34 pretreatment. LPS induced a notable increase in the protein PARylation level in mice lungs from 0.5 h and peaked at ∼1 h (Fig. 10a). This induction was blocked by the PJ34 pretreatment (Fig. 10b). Accordingly, the interaction of HuR with PARP1, as well as HuR's PARylation level, markedly increased after 1 h of LPS exposure, and the PARylation was also diminished by PJ34 administration (Fig. 10c). The results implied a potential physiopathological impact of PARP1 in modulating HuR's function in response to inflammatory stimulation.

## Discussion

PARP1's role in the regulation of gene expression under a variety of conditions has been well established. While a large number of studies have reported that PARP1 promotes gene transcription[28], our present work demonstrated that augmentation of the stability of pro-inflammatory mediator mRNAs presenting a regulatory mechanism of PARP1 in gene expression at the post-transcriptional level.

The 'steady-state' level of transcripts in eukaryotic cells is an outcome of the competition of RNA synthesis and degradation[32]. The best-studied instability elements in mammalian mRNA are the AREs[33]. Up to 8% of the genes in the human genome contain at least one putative ARE in their 3′-UTR[34]. The stability of ARE-containing mRNAs is mediated by ARE-binding proteins. Among them, the Elavl family members HuR, HuB, HuC and HuD stabilize target mRNAs and/or stimulate their translation[23]. In our present study, series dual-reporter assays (Fig. 1) suggested that HuR is the factor PARP1 acting to modulate the stability of ARE-containing mRNAs.

Our present study demonstrated an inflammatory stimulation-induced interaction of HuR with PARP1 and the subsequent PARylation of HuR. Recently, several groups' proteome-wide studies identified PARylation targets. A large number of PARylated substrates are involved in RNA-related metabolic processes[35,36] other than chromatin structure modulation, DNA repair, transcription and cell death. Intriguingly, HuR was identified as a PARylation target under $H_2O_2$ or methyl methane sulfonate stimulation, indicating the coordination of RNA metabolic processes in response to genotoxic stress[27].

The functional regulation of HuR relies on diverse post-translational modifications. To date, HuR has been identified as a substrate of serine and threonine phosphorylation by PKC, Chd2, p38 and Cdk1 (reviewed in ref. 37). However, emerging data also reported that other types of post-translational modifications, including tyrosine phosphorylation, methylation and ubiquitylation, either positively or negatively regulate the functions of HuR[38,39]. Our present study strikingly revealed that PARP1 binds to and PARylates HuR in cells upon LPS exposure (Figs 2, 3 and 10). Pull-down assays revealed that HNS and RRM3 are involved in the interaction with PARP1. In addition, *in vitro* PARylation assays showed HNS was strongly modified on D226 (Fig. 3). *In vivo*, the D226 mutation abolished the PARylation of HuR in cells challenged by LPS (Figs 8 and 9).

HuR's function is considered to be controlled in two principal ways: (1) being mobilized from the nucleus to the cytoplasm and (2) altering its association with target mRNAs[19].

So far, serine phosphorylation within HNS is the well-established mechanism regulating the nucleocytoplasmic shuttling of HuR. Cdk1 phosphorylates HuR at S202 during G2, thereby helping to retain it in the nucleus, in association with 14-3-3, and hindering its post-transcriptional function and anti-apoptotic influence[26]. The recruitment of HuR to the cytoplasm is enhanced by S221 phosphorylation, which is a consequence of the direct interaction of PKC-alpha with nuclear HuR in response to

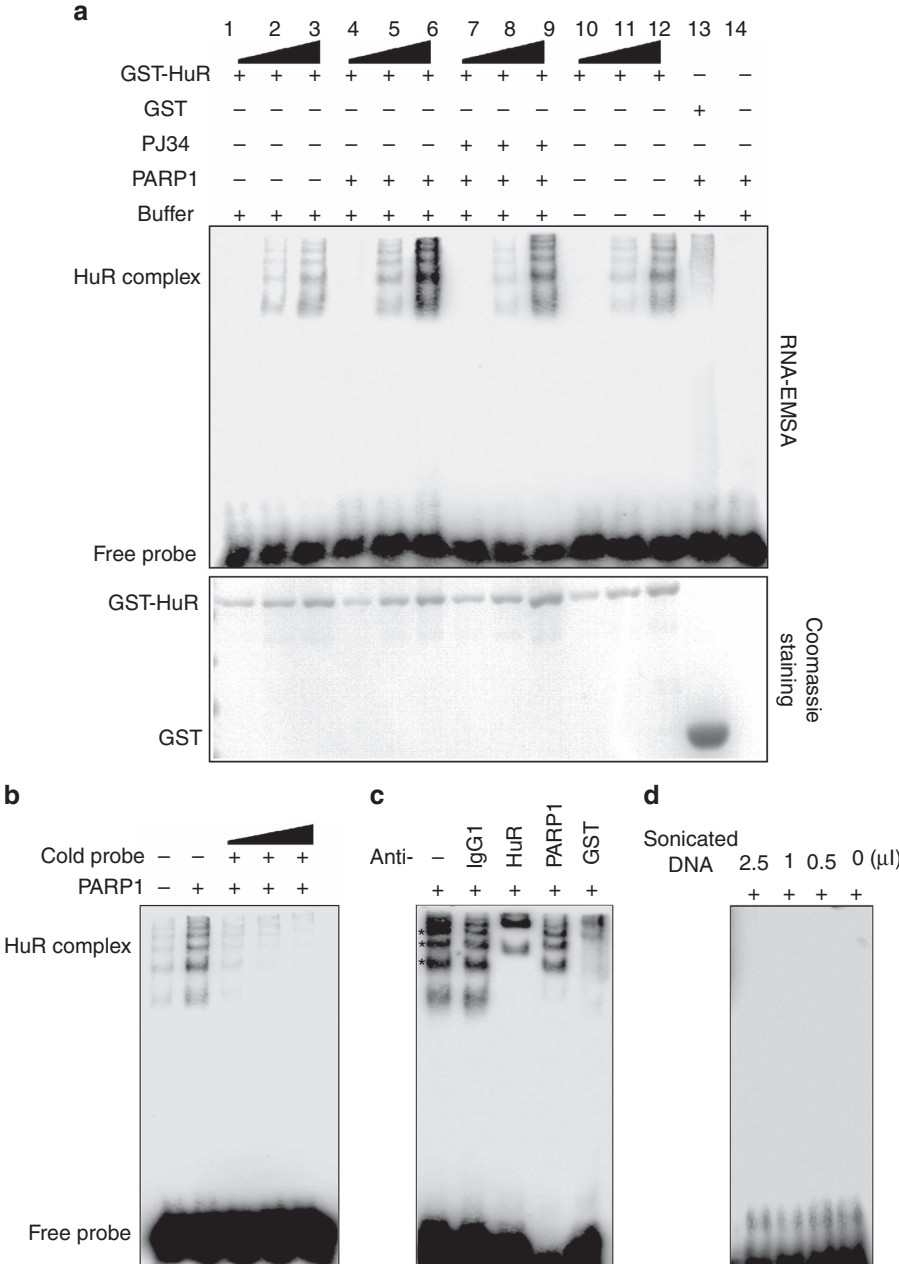

**Figure 6 | PARP1 increases the binding of HuR with ARE-containing RNA.** (**a**) Binding of the HuR with the ARE motif-containing RNA oligo is enhanced by PARylation. *E. coli*-expressing GST-HuR or GST was purified and eluted, and then subjected to PARylation, or not. Differently conditioned proteins were titrated (20-, 10- and 5-fold diluted, respectively) and then incubated with biotin-labelled tandem ARE repeat RNA oligos as indicated (upper panel). The gel retardation assay was performed to detect the binding of GST-HuR or GST to probes. The input amount of GST-HuR or GST within the binding system was visualized by Coomassie brilliant blue staining (lower panel). (**b**) Cold probe competition assay. GST-HuR was subjected to PARylation, or not, and then 10-fold diluted as indicated in lanes 2 and 5 in a. The diluted proteins were incubated with biotin-labelled probes in the presence or absence of 50-, 100- and 200-fold excessive non-labelled RNA. Gel retardation assays were performed. (**c**) Antibody supershift analysis. GST-HuR was subjected to PARylation and then 10-fold diluted as indicated in lane 5 in a. The diluted proteins were incubated with biotin-labelled probes in the presence of IgG1, or antibodies against HuR, PARP1 or GST, as indicated. Gel retardation assays were performed. (**d**) Sonicated DNA is not blocking the binding of PARP1 with the ARE motif-containing RNA oligo. Titrated sonicated DNA was utilized in PARP1 activation, and then ARE motif-containing RNA oligos were incubated with PARP1 in the binding buffer. Gel retardation assays were then performed.

increases in ATP or angiotensin II[40,41]. Our present study provided substantial evidence to show PARP1 is indispensable for the nucleocytoplasmic shuttling of HuR (Fig. 4). D226 mutation resulted in the handicapped nucleocytoplasmic shuttling of HuR in LPS-exposed macrophages (Figs 8 and 9).

The involvement of PARylation in protein nuclear export has been addressed previously. RelA/p65 PARylation decreased its interaction with chromosomal maintenance 1 (CRM1, also known as exportin 1) upon TLR4 stimulation, leading to NF-κB nuclear retention, which ultimately influenced NF-κB-dependent gene expression[42]. Also, PARP1-mediated PARylation of p53 blocked the interaction of p53 with CRM1, resulting in the nuclear accumulation of p53 (ref. 43). The export of HuR to the cytoplasm is regulated mainly in a CRM1-dependent manner

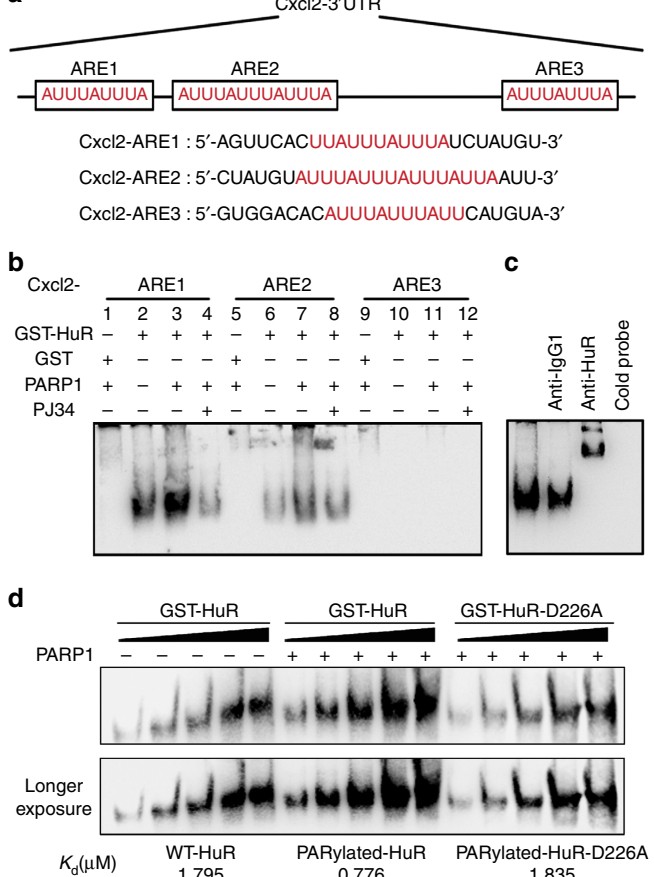

**Figure 7 | PARP1 increases the binding of HuR with AREs in *Cxcl2* mRNA's 3′-UTR.** (**a**) Illustration of three AREs in *Cxcl2* mRNA's 3′-UTR. (**b**) PARP1 activation promotes the association of HuR with ARE1 and ARE2 in *Cxcl2* mRNA's 3′-UTR. In lanes 1, 5 and 9, GST incubated with PARP1 was applied as a control; GST-HuR alone (lanes 2, 6 and 10), and GST-HuR incubated with PARP1 in absence (lanes 3, 7 and 11) or presence (lanes 4, 8 and 12) of PJ34 were subjected to a gel retardation assay. (**c**) Antibody supershift analysis and cold probe competition assay on ARE1 probe. (**d**) Incubation with PARP1 enhances the interaction of WT HuR but not D226A HuR with *Cxcl2*-ARE1-containing RNA. Varying concentrations of GST-HuR or GST-HuR-D226A (100, 200, 400, 800 and 1,600 nM; calculated by utilizing bovine serum albumin (BSA) concentration standard formula) were incubated with PARP1, or not, and a gel retardation assay was performed as described above. Quantifications of bound and unbound signal allowed dissociation constants ($K_d$) to be determined.

through its association with nuclear ligands pp32 and APRIL, which contain nuclear export signals that are recognized by the exported CRM1 (ref. 44). In addition, HuR serves as an adaptor for *c-fos* mRNA export through another pathway that involves the interaction of HNS with transportin 2 (ref. 44). Thus, whether the downstream pathways mediating nuclear export of PARylated HuR are CRM1-dependent or -independent requires further investigation.

Many other reports have focused on the binding of cytoplasmic HuR with target mRNA[25,45] to determine its function in regulating mRNA stability. Whereas we propose that PARylated HuR might bind to ARE-containing mRNA before the mRNA is transported to the cytoplasm since both PARP1 and HuR are located in the nuclei of quiescent cells, where the interactions of the two proteins and the PARylation of HuR are induced. The binding of HuR to transcripts may also affect other nuclear

events, such as splicing, polyadenylation, intracellular trafficking, translation and modulation of mRNA repression by miRNAs[37]. Thus, the influence of PARylation on the binding of HuR to target mRNA may have other profound effects downstream in different signalling pathways, which needs to be further explored in the future.

In addition, the inhibition of PARP1 or D226A mutations resulted in decreased binding of HuR to *Cxcl2* mRNA upon LPS stimulation (Figs 5–9). Recombinant protein RNA-EMSA assays showed that PARP1-inflicted modifications enhanced the binding of HuR to ARE-containing RNA oligos, indicating that PARylation of HuR increases its association with AREs, thereby regulating the stability of ARE-containing mRNA. RRM1 and RRM2 are considered the major domains to interact with RNA cargos. Previous studies have shown that phosphorylation at S88 in RRM1, T118 and S158 in RRM2, S100 between RRM1 and RRM2 affects HuR binding to numerous mRNAs[37]. Whereas a recent study also showed that the phosphorylation of S318 in RRM3 by PKCδ affects the binding of HuR to target mRNA[46]. It is somewhat surprising that the modification of D226, which is located in the HNS, affected the binding of HuR to target mRNA. Our combined data from GST pull-down and *in vitro* PARylation assays suggested a possible mechanism in which the PARylation of D226 in HNS might lead to a conformational change, facilitating the recognition of RRM(s) to the target mRNAs. In support, the Janus kinase 3 elicited the phosphorylation of Y200, an amino acid within HNS, and also influenced the binding of HuR to *SIRT1* mRNA[39]. However, the RNA-recognizing domains may not be PARylated (Fig. 3d) because highly negatively charged PAR chains may block the interaction with RNAs that are also negatively charged. Nevertheless, several RRM-containing proteins (for example, the heterogeneous nuclear ribonucleoprotein family[47,48] and the RNA processing factors NONO and RBMX[49,50]) have been demonstrated as PAR readers, interacting with PARylated proteins, which adds another regulatory possibility for protein PARylation in the binding of HuR with target RNA.

An intriguing study demonstrated that upon lethal stress, HuR undergoes caspase-mediated cleavage at D226 in the cytoplasm. This cleavage activity is associated with the apoptosome activator pp32/PHAP-I, and this caspase-mediated cleavage constitutes a regulatory step that contributes to an amplified apoptotic response[51]. Here we deduced D226-mediated PARylation may impair the effect of the apoptosome and further influence HuR functions by slowing down its turnover rate in the cytoplasm.

In addition, the role of protein PARylation in post-transcriptional regulation of gene expression may involve other members of the human PARP family (for example, PARP-5a, -12, -13.1, -13.2, -14 and -15) at multiple levels. These cytoplasm-located PARPs recruit and modify the ARE-binding proteins[52,53] or microRNA-binding Agos[54], directing mRNA-carrying complexes to stress-granules, blocking mRNA translation[28,55] or destabilizing target mRNA in an exosome-dependent manner[53].

HuR affects cell fate by regulating the stability and/or translation of mRNAs that encode proteins contributing to the vast majority of cellular processes, including cell growth and differentiation, metabolism, migration, immune response, apoptosis, and senescence[56]. The stabilization of the mRNAs encoding important inflammatory mediators[20,57] constitutes an important paradigm of HuR's functions. Many inflammatory mediator mRNAs known to be regulated at the stability level[58–61], and shown subjected to PARP1 regulation in the present study, are ARE-containing, such as *Il1β*, *Cxcl1*, *Cxcl2* and *Ccl11*. Moreover, although there is no study addressing its stability, *Il11* mRNA contains typical AREs in its 3′-UTR, and PARP1 inhibition

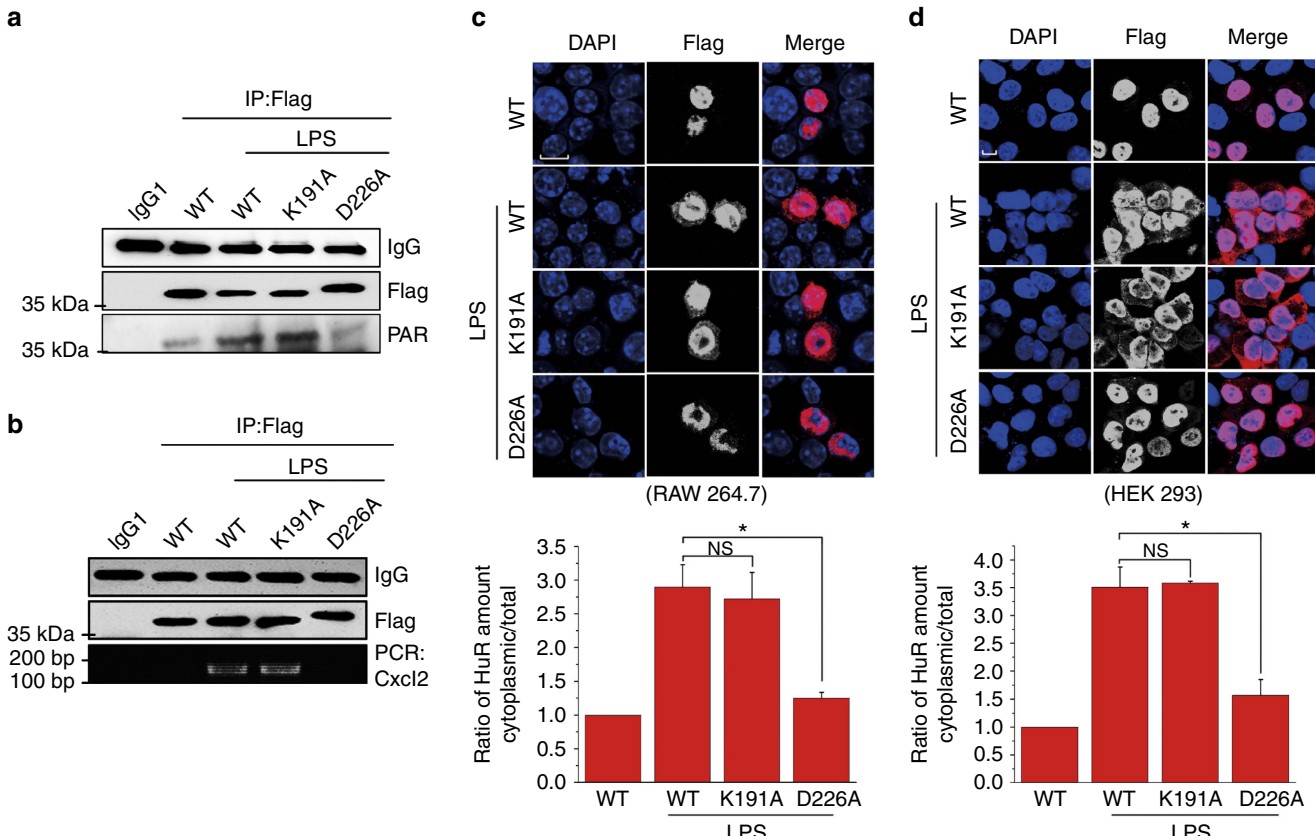

**Figure 8 | The HuR D226 mutant cannot be effectively PARylated, bind with *Cxcl2* mRNA, undergo nucleocytoplasmic shuttling. (a)** The HuR D226 mutant showed no PARylation in LPS-exposed cells. HEK 293 cells were transfected with wild-type (WT) Flag-HuR, as well as K191A and D226A mutant plasmids, and then challenged with LPS, or not for 5 h. Immuno-precipitates were prepared using a FLAG antibody and then subjected to immunoblotting to detect PARylation levels. **(b)** The HuR D226 mutant is defective in binding to *Cxcl2* mRNA in response to LPS exposure. HEK 293 cells were transfected with WT Flag-HuR, as well as K191A and D226A mutant plasmids. RNA-IP was conducted using a FLAG antibody and the levels of precipitated *Cxcl2* mRNA were detected by PCR with reverse transcription and electrophoresis as described in Methods. **(c,d)** The HuR D226 mutant failed to undergo nucleocytoplasmic shuttling in LPS-exposed cells. Experiments were undertaken as described above. Immune-fluorescence staining was conducted using a FLAG antibody to detect the location of WT HuR and the mutants in RAW 264.7 **(c)** and HEK 293 cells **(d)**. Scale bar, 10 μm. The cytoplasmic distribution of HuR was quantified by densitometry analysis using Image J software (version 1.44; lower panels) as described in Methods. Scale bar, 10 μm. Data were expressed as mean ± s.d. ($n = 5$), and analysed by one-way analysis of variance. *$P < 0.05$, NS, not significant.

resulted in a significant decrease in its mRNA stability as shown by plate-based real-time PCR arrays. The combined results implied that PARP1 may regulate the stability of a group of ARE-containing mRNAs by acting on the ARE-binding protein HuR.

In summary, our present study demonstrated that binding to PARP1 and PARylation are crucial for HuR-mediated mRNA stability, thereby uncovering a new mechanism to regulate gene expression at the post-transcriptional level. Our data also suggest a potential strategy to treat diseases closely linked to increased mRNA stability, such as inflammation-related disorders and cancers, through the inhibition of the PARylation of HuR.

## Methods

**Antibodies and reagent.** Monoclonal antibodies against PARP1 (1:2,000, B-10, sc-74470), HuR (1:2,000, 3A2, sc-5261), Histone H1 (1:1,000, AE-4, sc-8030), PARP2 (1:1,000, F-3, sc-393310), IRAK1 (1:2,000, B-5, sc-55530), rabbit p-IRAK1 (1:1,000, Ser 376, sc-325147) and goat polyclonal antibody TTP (1:1,000, N-18, sc-8458) were purchased from Santa Cruz Biotechnology (Santa Cruz, CA, USA). Anti-β-tubulin (1:8,000, HC101) and anti-β-actin (1:8,000, HC201) mouse monoclonal antibodies were purchased from TRANS (Beijing, China). Monoclonal antibody against PAR (1:2,000, ALX-804-220) and anti-PARP1 rabbit polyclonal antibody (1:5,000, ALX-210-302-R100) were from Alexis (San Diego, CA, USA). The anti-PAR rabbit polyclonal antibody (1:2,000, 4336-BPC-100) was from

Trevigen (Gaithersburg, MD, USA). The monoclonal antibody against FLAG (1:8,000, F1804) was from Sigma (Saint Louis, MO, USA). Protein synthesis inhibitor CHX (C1988), transcription inhibitor Act D (A1410), PARP1 inhibitor PJ34 (P4365), 3-AB (A0788), Ribonuclease A (R5503), PARG (SRP8023, 40 ng per 50 μl) and LPS (L2630) were from Sigma. Olaparib (AZD2281) from Selleckchem (Houston, TX, USA).

**Preparation of murine peritoneal macrophages.** pMφ cells were isolated from C57BL/6J mice as described previously[62,63]. Briefly, 20–22 g mice were injected with 2 ml of 4% thioglycollate. Two days after the injection, peritoneal exudate cells were isolated by washing the peritoneal cavity with ice cold PBS. Cells were incubated for 2 h, and non-adherent cells were removed. The macrophages were cultured with DMEM (Invitrogen, Carlsbad, CA, USA) containing 10% fetal bovine serum in Petri dishes for 3 days at 37 °C. More than 95% of the adherent cell population was that of macrophages, as determined by staining with monoclonal antibody F4/80 (ref. 64).

**Cell culture and treatment.** Murine RAW 264.7 macrophages and human embryonic kidney 293/hTLR4A-MD2-CD14 (HEK 293) cells stably transfected with the human *TLR4*, *MD2* and *CD14* genes (InvivoGen, San Diego, CA, USA). In the present study, for simplification, HEK 293 refers to this cell line. Cells were cultured in DMEM (Invitrogen) supplemented with 10% (v/v) fetal bovine serum and antibiotics. Mycoplasma contamination in the cell culture was negative detected by using CycleavePCR Mycoplasma Detection kit (Takara Bio Inc., Japan). For the immune challenge, the dose of LPS was 500 ng ml$^{-1}$. To inhibit *de novo* transcription, cells were treated with 10 μg ml$^{-1}$ of Act D. The dose of the PARP 1 inhibitor PJ34 was 2.5 μM as previously described[10,65]. To inhibit PARP1

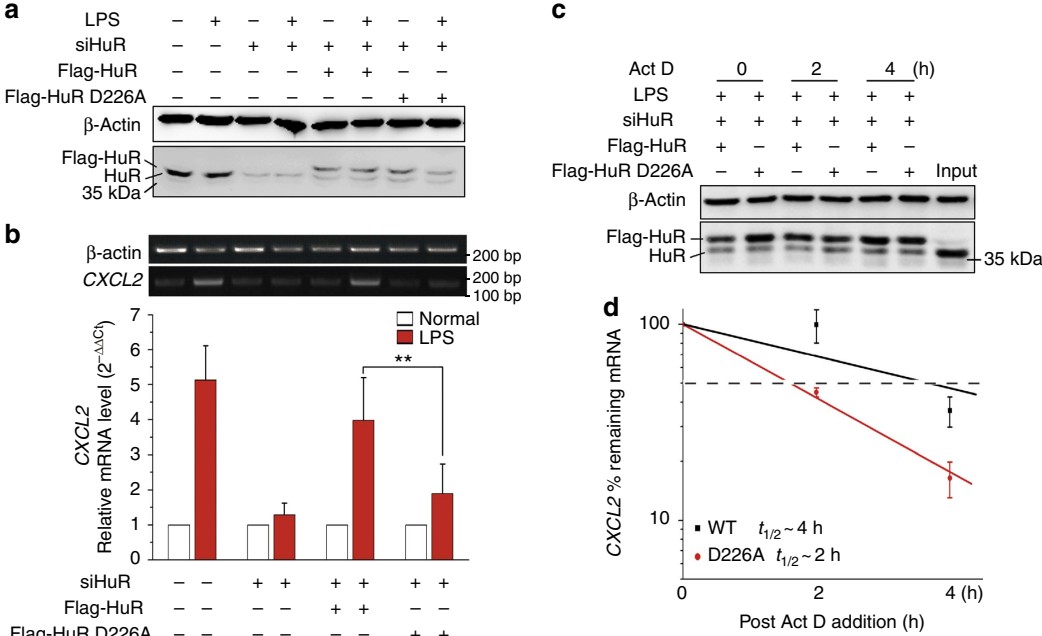

**Figure 9 | The HuR D226 mutant is incapable to sustain the *CXCL2* mRNA level. (a,b)** *CXCL2* mRNA expression is impaired in D226A HuR-expressing cells. Endogenous HuR in HEK 293 cells was silenced using siRNA-targeting a distinguished sequence of human HuR, and then murine WT HuR and D226A HuR expressional plasmids were transfected. Immunoblotting was performed to detect the interference of endogenous HuR, as well as the ectopic expression of Flag-tagged murine WT and D226A HuR (**a**). Cells expressing WT or D226A HuR were stimulated with LPS, or not for 2 h. *CXCL2* mRNA levels were detected by PCR with reverse transcription and electrophoresis (upper), or real-time PCR (lower) (**b**). (**c,d**) *CXCL2* mRNA's half-life is shortened in D226A HuR-expressing cells in response to LPS stimulation. The ectopic expression of murine WT HuR and D226A HuR in endogenous HuR-silenced HEK 293 cells was carried out and detected as described above. The last lane loaded with cell lysate from control cells serves as a marker (**c**). Cells expressing WT or D226A HuR were stimulated with LPS for 1 h to boost inflammatory gene expression and then subjected to transcriptional inhibition for different lengths of time (as indicated). Real-time PCR was performed to assess the remaining *CXCL2* mRNA levels. Half-lives of different samples are indicated in the inset. Data were expressed as mean ± s.d. ($n = 5$), and analysed by one-way analysis of variance. **$P < 0.01$.

activation, 3-AB and Olaparib were applied at 20 and 5 μM to the cell culture. The dose of Rnase A was 10 μg ml$^{-1}$. To inhibit protein synthesis, 10 μg ml$^{-1}$ of CHX was utilized[45], and 40 ng of PARG was added to the PARylation assay.

**Reverse transcription and real-time PCR.** Total RNA was extracted using TRIzol reagent (Invitrogen), and 1 μg of purified RNA from each sample was transcribed to complementary DNA. Primers for real-time PCR included: mCxcl2: forward: 5′-TCAATGCCTGAAGACCC-3′, reverse: 5′-TGGTTCTTCCGTTGAGG-3′; mCcr7: forward: 5′- GCGAGGACACGCTGAGAT-3′, reverse: 5′-GCCGATG AAGGCATACAA-3′; mGAPDH: forward: 5′-CTCATGACCACAGTCCATG C-3′, reverse: 5′-CACATTGGGGGGTAGGAACAC-3′; and mβ-actin: forward: 5′-AACAGTCCGCCTAGAAGCAC-3′, reverse: 5′-CGATGACATCCGTAA AGACC-3′.

**Stability of mRNA.** To measure the effect of PARP1 activity in regulating the stability of the inflammatory mediator mRNA, a classical approach is applied[14]. RAW 264.7 or pMφ cells were exposed to LPS for 1 h, and then transcription inhibitor Act D was added to the medium with or without the maintenance of LPS (± PJ34) for 0, 1, 2, 3 and 4 h. The level of mRNA was measured. Individual PCR amplification reactions were performed and analysed as described above. Plate-based inflammation-related cytokines and chemokines PCR arrays were used as suggested by the manufacturer (SABiosciences, Valencia, CA, USA).

**Constructs.** To construct reporter plasmids to detect the effects of the 3′-UTR on the stability of target gene mRNA, the *Cxcl2*-3′-UTR was amplified by PCR using murine complementary DNA as the template and was cloned into the vector pGL3-control (Promega, Madison, WI, USA). Plasmids Flag-HuR, GST and GST-HuR were kindly provided by Dr Myriam Gorospe (Laboratory of Cellular and Molecular Biology; National Institute on Aging, National Institutes of Health, USA). The domain mutations GST-HuR-RRM1, GST-HuR-RRM2, GST-HuR-HNS, GST-HuR-△RRM1, GST-HuR-△RRM1 + RRM2 and GST-HuR-RRM3 were developed from GST-HuR. A Fast Mutagenesis System kit (FM111, TRANS) was used to produce K191A and D226A site mutations in GST-HuR-HNS, GST-HuR and Flag-HuR.

**Luciferase assay.** RAW 264.7 cells were seeded in the 24-well plates overnight in the absence of antibiotics. The cells were then transfected with the *Cxcl2*-3′-UTR luciferase reporter plasmid, control vector (pGL3-Control) and Renilla luciferase reporter plasmid (an internal control, Promega) using Lipofectamine 2000 transfection reagent (Invitrogen). Cells were challenged with or without LPS 5 h later, and then lysed in 100 μl passive lysis buffer, and the extracts (20 μl) were analysed for luciferase activity using a Dual Luciferase Reporter Assay kit (Promega). To further analyse the effects of the 3′-UTR on mRNA levels of target genes, RNA was extracted and 1 U of DNase was used per 1 μg of RNA to eliminate plasmid DNA contamination. Firefly luciferase mRNA levels were measured by real-time PCR and calibrated to that of Renilla. Primers used were as follows: firefly luciferase: forward: 5′-GGTGGACATCACTTACGC-3′, reverse: 5′-CTCACGCAG GCAGTTCTA-3′; and Renilla luciferase: forward: 5′-AGCCAGTAGCGCGGT GTATT-3′, reverse: 5′-TCAAGTAACCTATAAGAACCATTACCAGATT-3′.

**siRNA.** siRNAs targeting murine PARP1 (#1: 5′-CCAUCAAGAAUGAA GGAAAUU-3′, #2: 5′-UUUCCUUCAUUCUUGAUGGUUUU-3′), murine HuR (#1: 5′-CAGAAACAUUUGAGCAUUGUA-3′, #2: 5′-ACUCGCCUGCU AGGCGGUUUGGA-3′) and human HuR (5′-UGCCGUCACCAAUGUGA AAGU-3′), and siPARP2 (sc-152028, Santa Cruz Biotechnology) were used at 100 pM. Cells were seeded in plates, incubated in growth medium without antibiotics overnight, and then transiently transfected with RNA oligos using lipofectamine 2000 following the manufacturer's instructions. At 4–6 h post transfection, cells were replaced with complete medium to promote recovery.

**Cell fractionation.** For immunoblotting, cells were lysed in lysis buffer[66] for 30 min on ice. Lysates were centrifuged at 12,000g for 20 min at 4 °C, and the supernatants were taken as the whole-cell extract. Cytoplasm and nuclear fractions were prepared using the CelLytic NuCLEAR Extraction kit (Sigma) following the manufacturer's guidance. Briefly, cells were lysed with cytosolic lysis buffer for 20 min, lysates were centrifuged (11,000g, 1 min, 4 °C), and supernatants were collected (CE). The pellets were washed twice with cytosolic lysis buffer and lysed with extraction buffer. Nuclear lysates were clarified by centrifugation (21,000g, 5 min, 4 °C), and the supernatants were collected (NE).

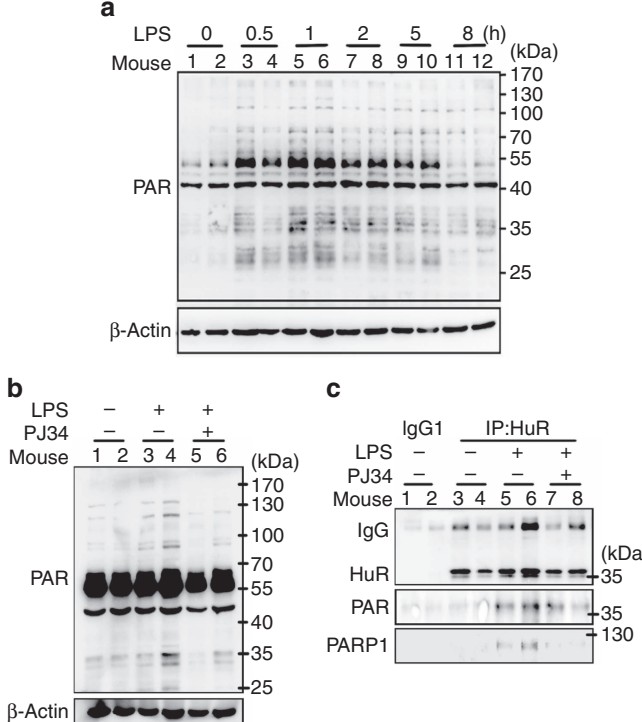

**Figure 10 | LPS induces association of PARP1 with and PARylation of HuR in mice lungs.** (**a**) LPS exposure increased protein PARylation. Mice were challenged with LPS (50 µg per mouse) through the intranasal route for different time intervals, and then the lungs were excised and homogenized. Immunoblotting was performed to detect the PARylation levels of protein. (**b**) Activation of PARP is attributed to the increased protein PARylation. Mice were challenged with LPS (50 µg per mouse) for 1 h, with or without intraperitoneal pretreatment of PJ34 (10 mg kg$^{-1}$). Lung homogenates were prepared, and immunoblotting was performed to detect protein PARylation levels. (**c**) PARP1 inactivation inhibited the interaction of HuR with PARP1 and the PARylation of HuR. Mice were treated as described in the legend to **b**. PARylation levels of HuR and the co-precipitated PARP1 in the HuR antibody-associated complex from lung homogenates were examined.

**Immunoblotting and immunoprecipitation.** RAW 264.7 cells were cultured, stimulated and lysed. WE, CE and NE were prepared as described above. Then, 20 µg of protein from each sample was resolved by SDS–polyacrylamide gel electrophoresis. To carry out immunoprecipitation, WE, CE or NE extracts were cleared with protein G beads for 1 h at 4 °C before being incubated with 4 µg of monoclonal anti-HuR, PAR, PARP1 and FLAG antibodies, or, alternatively, the same amount of IgG, overnight at 4 °C. After washing, the precipitated proteins were analysed by immunoblotting[66]. The un-cropped scans of some important blots are supplied as Supplementary Fig. 8 in the Supplementary Information.

**GST-fused protein purification and GST pull-down assay.** GST and GST-fused proteins were expressed in *Escherichia coli* strain BL21. The induction was performed by adding 1 mM isopropyl-β-D-thiogalactopyranoside to an OD 1.0 culture at 37 °C for ∼2–3 h. Whole bacteria lysates were applied to glutathione Sepharose 4B (GE Healthcare Life Science, Uppsala, Sweden), and GST-tagged proteins were purified according to the manufacturer's instructions. For pull-down experiments, GST and GST-fused proteins immobilized on 40 µl of Glutathione Sepharose 4B were incubated with 1 ml of cell extract at 4 °C for ∼1–3 h. After three washes with Nonidet P-40 lysis buffer, the bound proteins were analysed by immunoblotting.

***In vitro* PARylation assay.** A GST-fused protein *in vitro* PARylation assay was set up by modifying the method provided by the HT Universal Chemiluminescent PARP Assay kit (Trevigen, Gaithersburg, MD, USA). Briefly, GST and GST-fused proteins immobilized on 25 µl of glutathione Sepharose 4B were incubated with recombinant PARP enzyme and PARP cocktail at room temperature for 1 h. After three washes with Nonidet P-40 lysis buffer, the bound proteins were analysed by immunoblotting.

**Immunofluorescence microscopy.** Cells were fixed with 10% (v/v) formaldehyde, permeabilized with 0.5% (v/v) Triton X-100, blocked with 2% (w/v) bovine serum albumin and incubated with primary antibodies recognizing HuR (1:200), PARP1 (1:200), FLAG (1:500), TTP (1:100) or PAR (1:200). Secondary antibodies were used to detect primary antibody–antigen complexes with different colour combinations as needed. The nuclei of the cells were stained with 4,6-diamidino-2-phenylindole for 5 min. Images were acquired using a confocal microscope (Nikon, Tokyo, Japan).

To quantify the nuclear-cytoplasmic redistribution of HuR, densitometry analysis was conducted by using Image J software (version 1.44). The total HuR amount was measured first, and then that of nuclear HuR, thereby we had the cytoplasmic amount of HuR by taking nuclear amount of HuR away from that of the total. The redistribution of HuR was estimated by dividing the cytoplasmic amount of HuR by that of total.

**RNA-IP.** RAW 264.7 cells were exposed to 500 ng ml$^{-1}$ LPS for 1 h to boost pro-inflammatory gene expression, followed by the addition of Act D. In addition, the cells were withdrawn from LPS, maintained in LPS during incubation or treated with LPS plus PJ34 for another 2 h. Then, WE and CE were pre-cleared and immune-precipitated using protein G agarose/salmon coated with an anti-HuR antibody or alternatively, the same amount of IgG (4 µg). One-half of the bead–antibody–protein/mRNA-bound complexes for each sample were washed three times and analysed by immunoblotting, and the other half was washed and used for mRNA isolation. The mRNA was isolated using an RNA sample total RNA kit (TIANGEN, Beijing, China). The level of mRNA was measured by quantitative PCR as described above. The precipitated RNA target was analysed by dividing the amount of RNA in the IP by that in the input. To determine the stability of the mRNA in the HEK 293 cells, human *Cxcl2* primers were applied: forward: 5′-CAAACCGAAGTCATAGCC-3′, reverse: 5′-GAACAGCCACCA ATAAGC-3′.

**RNA-EMSA.** GST or GST-HuR proteins were induced from an 8 ml *E. coli* culture with an OD 1.0 and then extracted using 70 µl of glutathione Sepharose 4B beads. The recommended proteins were purified and eluted in 100 µl buffer (50 mM Tris–HCl (pH > 8.0), 100 mM KCl and 40 mM glutathione). Then, 25 µl of the eluted proteins were incubated with or without PARPas describe previously. To perform RNA-EMSA and supershift analyses, a Chemiluminescent RNA EMSA kit (20158, Thermo Fisher Scientific, Waltham, MA USA) was used. Briefly, titrated proteins (with or without PARylation) were dissolved in the EMSA interaction buffer (3 mM MgCl$_2$, 40 mM KCl, 5% glycerol, 2 mM DTT, 2 µg tRNA) and incubated with 20 µM of 5′ biotin-labelled RNA oligos for 40 min at room temperature. For supershift assays, 0.4 µg of specific antibodies or IgG were added to the mixture after 15 min of incubation at room temperature. The reaction mix was then loaded on to a 6% acrylamide native gel. RNA oligo probes utilized in the present study included: AU-rich RNA oligo: 5′-AUUUAUUUAUUUAUUUAUU UAUUUA-3′; Cxcl2-ARE1 RNA oligo: 5′-AGUUCACUUAUUUAUUUAUCU AUGU-3′; Cxcl2-ARE2 RNA oligo: 5′-CUAUGUAUUUAUUUAUUUAUUA AUU-3′; Cxcl2-ARE3 RNA oligo: 5′-GUGGACACAUUUAUUUAUUCAUGUA-3′. To compare the affinities of WT and D226A mutant HuR with the Cxcl2-ARE1 RNA oligo, varying concentrations of GST-HuR or GST-HuR-D226A were incubated with PARP1, or not, followed by interactions with 20 µM of 5′-biotin-labelled Cxcl2-ARE1 RNA oligo. A gel-shift was performed as described above. Band intensities were quantified, and $K_d$ values were determined as described previously ($K_d$ value = [protein][RNA oligo]/[complex])[31].

**Mouse work.** Six- to eight-week-old female C57BL/6 mice (20–25 g) were purchased from Jilin University (Changchun, Jilin, China). Mice were housed in a specific pathogen-free facility at NENU (Changchun, Jilin, China) and allowed unlimited access to sterilized feed and water. They were maintained at 23 ± 1 °C and kept under a 12-h light/dark cycle. All experiments were conducted in accordance with the Chinese Council on Animal Care Guidelines.

Mice were anaesthetized with pelltobarbitalum natricum (6.5 mg kg$^{-1}$), then randomized to be challenged with LPS (50 µg per mouse in 30 µl saline) or not, using the intranasal route[67], with or without an intraperitoneal pretreatment of PJ34 (10 mg kg$^{-1}$) 30 min before the LPS challenge[68]. Mice lungs were collected, and homogenates were prepared.

**Statistical analysis.** All experiments were performed at least three times for each determination. Data were expressed as means ± s.d.'s ($n = 5$) and analysed by one-way or two-way analyses of variance. The level of significance was accepted at $^*P < 0.05$, $^{**}P < 0.01$ and $^{***}P < 0.001$.

**Data availability.** All relevant data are available from the authors on request and/or are included with the manuscript (as Supplementary Information files).

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

## Acknowledgements

This work was supported by grants from the National Nature Science Foundation of China (grant nos 31371293 and 31571339 to X.B.), the Programme for Introducing Talent to Universities (grant no. B07017 to X.B. and X.Z.), NIH/NIEHS RO1 ES018948 (I.B.) and NIH/NIAID/AI062885 (I.B.). We thank Mardelle Susman for editing this manuscript (Department of Microbiology and Immunology, University of Texas Medical Branch at Galveston TX, USA).

## Author contributions

X.B., I.B. and X.Z. conceived and designed the experiments, Y.K., Y.H., X.G., J.W., K.W., X.J. and X.T. performed the experiments, Y.K., X.B. and X.Z. analysed the data, Y.K. and X.B. wrote the manuscript.

## Additional information

**Competing financial interests:** The authors declare no competing financial interests.

DOI: 10.1038/ncomms15191 **OPEN**

# Erratum: PARP1 promotes gene expression at the post-transcriptional level by modulating the RNA-binding protein HuR

Yueshuang Ke, Yanlong Han, Xiaolan Guo, Jitao Wen, Ke Wang, Xue Jiang, Xue Tian, Xueqing Ba, Istvan Boldogh & Xianlu Zeng

*Nature Communications* 8:14632 doi: 10.1038/ncomms14632 (2017); Published 8 Mar 2017; Updated 31 Mar 2017

The original version of this Article contained an error in the spelling of post-transcriptional in the title of the paper. This has now been corrected in both the PDF and HTML versions of the Article.

