## [Peer Review File · Nature Communications]

Reviewers' Comments:

Reviewer #1 (Remarks to the Author)

In the manuscript "PARP1 promotes gene expression at posttranscriptional level via modulating RNA-binding protein HuR" by Ke and colleagues, the authors report how PARylation of the RNA-stabilizing factor HuR can affect the stability of mRNA upon LPS treatment in cell culture models. This is solid work, most experiments were designed carefully, and most of the data are presented adequately. Also, the paper is well written. My major concern is related to the potential impact of the study, which largely focuses on the analysis of the interaction of PARP1 and HuR with regards to Cxcl2 mRNA stability, only. In the present form the paper suits well for a journal with an impact factor of ~5. To increase the impact of the manuscript it will be necessary to provide more evidence that the post-transcriptional regulation of mRNA levels by HuR and PARylation is a general mechanism upon inflammatory stimuli, to define its specificity, and to highlight its pathophysiological importance in an in vivo model.

General comments:

With the exception of Figure S1 the authors exclusively study the stability of Cxcl2 mRNA. Firstly, it is not clear to the reviewer why the authors exclusively focused on the Cxcl2 mRNA, since it seemed not to be included in the qPCR array analysis. Secondly, the authors should provide experimental data to explain why only a subset of mRNAs is sensitive to PARP inhibitor treatment. Thirdly, to prove that the regulation of HuR by PARylation is a general mechanism the authors need to study more than one mRNA species in further detail. E.g., it would be informative to conduct qPCR array analyses with a PARylation-deficient HuR mutant.

As the authors state in the manuscript, there is a strong functional interaction between PARP1 and NF- κ B-mediated gene expression. To what extent affects the NF- κ B pathway the results presented in the manuscript? Are the mechanisms completely independent or is there a partial overlap? Which of the PARP-inhibitor sensitive mRNA expression levels of Fig. S1e are related to the PARP1-NF- κ B and which to the PARP1-HuR regulation. And why is this the case?

At least one experiment should be included showing the in vivo relevance of the mechanism discovered, preferably using a mouse model.

It is not recommended any more to use PJ34 for PARylation research, since this inhibitor has been shown to elicit considerable off-target effects. It is advised to use inexpensive, more specific PARP inhibitors, such as olaparib and veliparib.

Throughout the manuscript, PAR immuno-blots should be presented completely (whole membranes), since PARylation affects proteins of variable size. Molecular weight makers should be included.

Covalent PARylation of HuR should be verified by mass spectrometry.

It would be interesting to know if HuR functions also as a reader of PARylation, i.e., does HuR bind PAR in a non-covalent manner via specific PAR binding motifs?

The PARylation dependent shuttling mechanism should be analyzed in more detail, e.g., is it dependent on CRM1, as previously shown for p53?

Specific comments:

Fig. 1:

Panel a: Statistical significance could be tested via 2-way ANOVA analysis

Panel h: A PARP1 immunoblot should be included

Fig. 2:

Panel a: Molecular weight marker is missing

Panel b and e: A densitometric analysis of independent replicates should be included

Fig.3:

Panel c: Why is automodified PARP1 protein not visible in the anti-PAR blot?

Fig.4: It is absolutely essential to quantify nuclear-cytoplasmic shuttling densitometrically via an

imaging software (e.g., Image J), since the effect is hardly visible from the pictures shown. Quantitative data should be included from independent replicates.

Fig.6: The influence of free PAR polymer on HuR-RNA interaction should be analyzed

Reviewer #2 (Remarks to the Author)

The work in this manuscript is a comprehensive analysis of a novel role of PARP1 in post-translational regulation of the RNA-binding protein, HuR. The results and their analysis and interpretation convincingly demonstrate that an association between PARP1 and HuR, followed by PARylation of HuR, is indispensable in regulating shuttling of HuR from the nucleus to the cytoplasm, a process essential to its activation and function. The authors further show that PARP1 regulation of HuR affects the expression of several pro-inflammatory genes, in particular Cxcl2. The Discussion does an excellent job in putting these new findings into the context of what is known about the transcriptional regulation and the functions of HuR. The implications of the findings for the development of new therapeutics for the treatment of inflammation-related disorders and cancer are also mentioned and are of special interest to investigators seeking more effective therapies based on disruption of HuR function.

While the experiments are well-designed and technically well-done, there are issues with the presentation of data. In particular, Figures 2 (RRM1, panel d; and panels e and f) and 6 (the whole figure) are messed up. The problems in Figure 6 do not allow for review of the data. The text in latter part of the Results section (from the sub-section "PARP1 enhances the binding of HuR to ARE-containing RNA" onward) is hard to follow, partly because of Figure 6 problems, but also because of syntax and wording.

Points that would strengthen the paper are:

- 1) Explain why PARG was used to inhibit PARP1 (at least provide a reference).
- 2) D226 is the cleavage site in HuR between CP1 and CP2. It is interesting that D226 is also the amino acid that, when mutated, disrupts HuR PARylation by PARP1. This coincidence and its possible implications should be discussed.
- 3) Does HuR shuttle from nucleus to cytoplasm in the D226A mutant?

Other minor points:

- 1) Line 223 - states that there is a moderate increase in PARylated HuR in the NE fraction. It appears to be about the same increase as in CE
- 2) Line 243 - There is no strong smear at the top of lane 3. Smear only present in lanes 6, 8, and 10.
- 3) The figure legends tend to be excessively long and repeat information in the text.

Reviewer #3 (Remarks to the Author)

In this manuscript, Ke et al. show that Poly(ADP-ribosyl)ation is required for proper cytokine expression upon immune-stimulation of macrophages. Specifically, they show that several cytokine mRNAs (including Cxcl2) are not stabilized when PARP1 is inhibited. They provide evidence that the RNA-binding protein HuR is PARylated upon immune-stimulation, which causes enhanced export of HuR into the cytoplasm as well as enhanced binding of HuR to RNA, which in turn allows HuR to stabilize Cxcl2 mRNA. This is an important finding for the field of posttranscriptional gene regulation and provides a novel mechanism by which cytokine mRNA stability is controlled.

While I am generally positive about this manuscript, there are several aspects the authors need to take care of. In particular, the HuR localization results need to be depicted more convincingly, and quantified thoroughly. Also, the RNA-IP experiments need to be normalized more stringently, otherwise one cannot draw conclusions about differences in the RNA-binding activity of HuR.

Specific comments:

L. 152: There is literature on the destabilizing activity of Cxcl2 3'UTR, which the authors need to quote. E.g. Numahata et al. J Cell Biochem 2003 90:976.

L. 158: The authors do not show that the ARE mediates the PAPR1 effect, they can only conclude that the Cxcl2 3'UTR mediates the effect.

L. 162: Ref. 21 is not appropriate for HuR as an ARE-mRNA stabilizing protein. The first reports were published by the Steiz and Shyu labs in 1998, both in EMBO Journal.

The authors frequently use descriptive wording when reporting on the effects observed. E.g. "severely impaired" (L.149), "significantly in/decreased" (L.151, 155), "markedly diminished" (L.157), and many more examples throughout the text. Instead, the authors should provide exact numbers, e.g. 20-fold increase, 3.4-fold decrease etc.

Fig. 1f shows only a relatively small effect of LPS on Cxcl2 3'UTR reporter mRNA levels (less than two-fold). The authors should confirm by Act.D chase experiments (as in Fig.1a,b) that the Cxcl2 3'UTR reporter mRNA is indeed stabilized by LPS treatment. Since this is a qPCR after transient transfection of the reporter plasmid, the authors should document that the signal they measure is derived from RNA, and not a plasmid DNA contamination in the RNA preparation. This can be done by a minus-RT control.

Fig. 1g: The data in panels 1e+f suggest that PARP1-dependent mRNA stabilization is responsible for a less than 2-fold increase in Cxcl2 reporter mRNA levels. In contrast, kd of HuR diminished Cxcl2 mRNAs levels about 6-fold in panel 1g. Does this mean that HuR also has an effect on transcription of Cxcl2?

L. 186: I do not understand why "The involvement of whole cell lysate input and molecular weight standard" should indicate "that the interaction of two molecules relies on their 188 full-length forms".

L. 200: The term "abundantly pulled down" is not warranted, since binding of HNS and RRM3 is clearly below binding of WT HuR (Fig. 2e).

Fig. 3a: The amount of HuR in the PAR-IP (right panel) is barely above the background level. The authors should either omit this experiment, or show a thorough quantification.

Fig. 4: In many of the IFs, the cytoplasmic localization of HuR upon LPS treatment is difficult to see. For one thing, the authors should depict the individual signals in black-and-white, which enhances the contrast, and use colors only in the merged panels. Second, it would be helpful to show larger magnifications, and increase the size of the image depicted. Third, and most importantly, the authors need to quantify the proportion of cytoplasmic HuR in the IFs by image analysis, and provide numbers in addition to the merely descriptive images. This comment also applies to Fig. 7c and d.

Fig. S5: I am not convinced by this experiment since the amount of overexpressed Flag-HuR is way below the level of endogenous HuR, and Flag-HuR cannot even be seen in the PARP1 IP. I suggest to omit this experiment.

Fig. 5: From these RNA-IPs, the authors cannot conclude that the interaction of HuR with Cxcl2 mRNA changes in an LPS- and PARP-dependent manner, because the Cxcl2 mRNA levels (input) change in a similar manner, as shown in Fig. 1g. To assess binding activity, the authors need to normalize the RNA signal in the HuR-IP to the amount of Cxcl2 mRNA in the input, and, to be fully accurate, also to the amount of HuR in the IP. This comment also applies to Fig. 7b.

Fig. 6a: While the data indicate enhanced binding of ARE-RNA to PARylated HuR compared to

unmodified HuR, the authors should take their EMSAs a step further and calculate K_d values for both forms of HuR. For this, they will need to increase the concentration of HuR until they reach saturated binding. Moreover, it would be interesting to know the K_d of the D226A mutant (in presence of PARP1).

Fig. 6f: According to the scheme in 6e, ARE1 and ARE3 are identical. Why is binding of the two oligos to HuR so different in 6f? The authors should depict the actual oligo sequences that were used in the EMSAs.

L. 372: The authors can only talk about Cxcl2 mRNA, not about mRNAs from pro-inflammatory genes in general, since they only measured Cxcl2 mRNA in Fig. 7f.

The authors should make an effort to improve English style and language of their manuscript, especially in the Results section.

Reviewer #1

We thank the reviewer for giving us the opportunity to improve our work and revise the manuscript. The comments and suggestions of the reviewer are constructive which helped us to significantly improve the quality of the manuscript, and to get further insight in the scientific issue how PARP1 regulate gene expression at posttranscriptional level via acting on HuR.

General comments:

With the exception of Figure S1 the authors exclusively study the stability of Cxcl2 mRNA. Firstly, it is not clear to the reviewer why the authors exclusively focused on the Cxcl2 mRNA, since it seemed not to be included in the qPCR array analysis. Secondly, the authors should provide experimental data to explain why only a subset of mRNAs is sensitive to PARP inhibitor treatment. Thirdly, to prove that the authors need to study more than one mRNA species in further detail. E.g., it would be informative to conduct qPCR array analyses with a PARylation-deficient HuR mutant.

Reply:

Yes, Cxcl2 is not included in the qPCR array analysis. We chose Cxcl2 mRNA as a target because murine Cxcl1 and Cxcl2 genes are highly conserved, their locations are closely clustered (chromosome 5) and their transcription is highly linked. Throughput of the qPCR array limits the selection of various inflammatory mediators, thus, maybe due to the consistence of their function and expression, one of them, Cxcl1 is chosen as representative for neutrophil chemotactic factor applied to plate array. However, we found that regarding the posttranscriptional level, the regulation of Cxcl2 is markedly higher than that of Cxcl1 (Fig. S1 F) might due to Cxcl2 mRNA 3'UTR contains more typical AREs, so we decided to use Cxcl2 mRNA as the paradigm in the present study.

Regarding “a subset of mRNAs is sensitive to PARP inhibitor treatment”, we drew the conclusion in the DISCUSSION that “The combined results implied that PARP1 may regulate the stability of a group of ARE-containing mRNAs by acting on the ARE-binding protein HuR”.

To support the conclusion, we followed the reviewer's suggestion, performed RNA-IP and examined the levels of a set of ARE-containing mRNA e.g. Cxcl1, Cxcl13 and IL-1 β in HuR's immuno-precipitates; the result showed LPS-induced association of HuR with these mRNA was significantly reduced upon addition of PARP1 inhibitor. We supplemented this data as Figure S5 in the revised manuscript.

Fig.S5

PARP1 inhibition blocked the association of HuR with ARE-containing mRNA mRNA

To prove that the regulation of HuR's function by PARP1 is a general mechanism, we did not conduct qPCR array analyses using a PARylation-deficient HuR mutant since RNA-IP assays already concluded PARP1 does not affect, and HuR does not bind with those mRNA with no AREs.

To address the reviewer's concern, we conducted ectopic expression of murine WT HuR and D226A HuR in endogenous HuR-silenced HEK 293 cells, in addition of measuring the half-life of remaining *CXCL2* mRNA (Fig. 7 h), we also examined other ARE-containing mRNAs such as *CXCL1*, *IL-8* and *TNFα*. The result showed the half-lives of remaining mRNA of these genes in WT HuR-expressing cells was ≥ 4 h; whereas, that was reduced to about 2 to 2.5 h in D226A HuR-expressing cells. Data were presented as **Figure S 6**. The results suggested regulation of HuR by PARylation is a general mechanism to control gene expression.

Fig.S6

Half-lives of ARE-containing mRNAs are decreased in D226A HuR-expressing cells in response to LPS exposure

The related depiction of these supplementary data was integrated in the DISCUSSION in the revised manuscript shown as below:

Many inflammatory mediator mRNAs known as regulated at the stability level⁵⁸⁻⁶¹, and shown subjected to PARP1 regulation in the present study, are ARE-containing, such as *Il1β*, *Cxcl1*, *Cxcl2* and *Ccl11*. Moreover, although there is no study addressing its stability, *Il11* mRNA contains typical AREs in its 3'UTR, and PARP1 inhibition resulted in a

significant decrease in its mRNA stability as shown by plate-based real time PCR arrays. The combined results implied that PARP1 may regulate the stability of a group of ARE-containing mRNAs by acting on the ARE-binding protein HuR. **To support, the levels of a set of ARE-containing mRNA e.g. *Cxcl1*, *Cxcl13* and *IL-1 β* in HuR's immuno-precipitates from LPS-treated RAW 264.7 cells were significantly reduced upon addition of PARP1 inhibitor (Figure S5). Also, stability of *CXCL1*, *IL-8* and *TNF α* mRNA was significantly lower in D226A mutant-expressing HEK293 cells than that in WT HuR-expressing ones (Figure S6).**

As the authors state in the manuscript, there is a strong functional interaction between PARP1 and NF-kB-mediated gene expression. To what extent affects the NF-kB pathway the results presented in the manuscript? Are the mechanisms completely independent or is there a partial overlap? Which of the PARP-inhibitor sensitive mRNA expression levels of Fig. S1e are related to the PARP1-NF-kB and which to the PARP1-HuR regulation. And why is this the case?

Reply:

It has been well acknowledged that PARP1 affects NF-kB-mediated transcription activation of proinflammatory mediators. However, in the present study, function of PARP1 mediated by NF-kB at transcriptional level was separated. As depicted in Figure S1 A, after 1 h of immune stimulation to achieve the transcription activation and the boost of increase in mRNA levels, transcription inhibition was inflicted to all samples. Levels mRNA in samples withdrawn, with maintenance of LPS as well as in presence of LPS and PARP1 inhibitor reflect the difference of mRNA stability in various samples. The mechanism we presented here is completely independent from the pathway mediated by NF-kB at transcriptional level.

At least one experiment should be included showing the in vivo relevance of the mechanism discovered, preferably using a mouse model.

Reply:

Following reviewer's suggestion, we set up a mouse lung inflammation model: Six to eight week-old female C57BL/6 mice (20-25g) were challenged by LPS (50 ug per mouse⁶⁷) via intranasal route as previously described. Unlike cell culture model, the in vivo animal model has the limitation to separate posttranscriptional changes of mRNA from that induced at transcriptional level. Thus, we tried to examine the interaction between PARP1 and HuR as well as HuR's PARylation post LPS exposure. The results were presented as Figure 8, and the depiction of the results in the revised manuscript (highlighted) is as below:

LPS induces interaction of PARP1 with and PARylation of HuR in mice lungs.
To investigate the inducible interaction of PARP1 with and PARylation of HuR in an *in vivo*

scenario, mice lungs were exposed to LPS via intranasal route with or without PJ34 pretreatment. LPS induced notable increase in protein PARylation level in mice lungs from 0.5 h and reaching the summit at about 1 h (Fig. 8a), which was blocked by PJ34 pretreatment (Fig. 8b); accordingly, the interaction of HuR and PARP1 as well as HuR's PARylation level markedly increased after 1 h of LPS exposure, which also was diminished by PJ 34 administration (Fig. 8c). Results implied the potential physio-pathological impact of PARP1 in modulating HuR's function in response to inflammatory stimulation.

Fig.8 LPS induces interaction of PARP1 with and PARylation of HuR in mice lungs.

(a) LPS exposure increased protein PARylation in mice lungs. Mice lungs were challenged with LPS (50 ug per mouse) via intranasal route for different time intervals, and then were excised homogenized. Immuno-blotting was performed to detect the PARylation levels of protein in lung tissue homogenates. (b) Activation of PARP is attributed to the increased protein PARylation in mice lungs. Mice lungs were challenged with LPS (50 ug per mouse) for 1 h with or without intraperitoneal pretreatment of PJ34 (10mg/kg). Lung homogenates were prepared; immuno-blotting was performed to detect protein PARylation levels. (c) PARP1 inactivation inhibited the interaction of HuR with PARP1 and PARylation of HuR in mice lungs. Mice were treated as described in the legend above (to b). PARylation levels of HuR and co-precipitated PARP1 in HuR antibody-associated complex from lung homogenates were examined.

It is not recommended any more to use PJ34 for PARylation research, since this inhibitor has been shown to elicit considerable off-target effects. It is advised to use inexpensive, more specific PARP inhibitors, such as olaparib and veliparib.

Reply:

Since we started the project several years ago when PJ34 were broadly applied in research, we have the limitation to re-do all the work using Olaparib instead of PJ34. Thus, we selected the key experiment, examined the effect of Olaparib on Cxcl2 mRNA stability (Figure S2a). We also addressed and excluded one of the off-target effects such as unspecific block of TLR/inflammasome signaling (Figure S2b). At this moment, we ask for an overlook from the reviewer on this inhibitor issue.

Throughout the manuscript, PAR immuno-blots should be presented completely (whole membranes), since PARylation affects proteins of variable size. Molecular weight makers should be included.

Reply:

As reviewer concerned, we addressed the whole-membrane PAR-immunoblot issue in our manuscript.

In the RESULTS section, when we depicted PARylation of HuR, we stated “As we expected, HuR was PARylated in extracts of LPS-exposed cells (Fig. 3a). Full membranes with molecular weight standards are shown in Figure S3b. Along with the absence of severe DNA damage, the length of the PAR polymer is considerably shorter, ranging from single residue to oligo units⁶. PARylated HuR did not exhibit apparent shift retardation, which was also noticed with other PARylated mRNA metabolism-related proteins²⁷”

Figure S3b showed HuR and PARylated HuR with molecular weight, which indicated 1) the specificity of HuR antibody and PARylation signal; 2) no apparent molecular weight shift occurred with PARylated HuR.

Thus we considered to keep PARylated HuR immuno-blots as currently they were in the main text of the manuscript to avoid taking too much space.

Covalent PARylation of HuR should be verified by mass spectrometry.

Reply:

We fully agree with the essentiality of the reviewer’s suggestion.

Despite of the company’s continuing claiming that without informing them a accurate molecular weight alteration of the particular modification, they could not analyze the mass spectrometry data, we still prompted them to try our chance to see whether there would be 541.0611, 1080.1222,such molecular weight increase of modification occurred on HuR’s D226.

We sent the in-gel sample. The company did trypsin digestion (20hr), followed by mass spectrometry and Mascot software analyses.

Quality control analyses revealed the enrichment and the fragmentation of peptides are good. And the peptides that had been successfully read out are shown as below:

RT: 0.00 - 60.01

DANLYISGLPR
DVEDM&FSR
DVEDMFSR
FAANPNQNK
ILQVSFK
NVALLSQLYHSPAR
VAGHSLGYGFVNYVTAK
VIRDFNTNK
VSYARPSSEVIK
AINTLNGLR
DFNTNK
DFNTNKCK
DKVAGHSLGYGFVNYVTAK
FGGPVHHQAQR
GVAFIR
IINSR
LGDKILQVSFK
LQSKTIK
RFGGPVHHQAQR
SLFSSIGEVESAK
TM&TQKDVEDM&FSR
TNLIVNYLPQNM&TQDELR
VLVDQTTGLSR

However, aligning the reads (highlighted in yellow) with HuR's amino acid sequence (coverage is 55.52%), we found that D226 (in red) is unfortunately not covered (the HNS domain is underlined). Another potential PARylated site in HNS domain, K191 (in green)

was within the peptide that had been read out, but the software analyses did not show modification occurrence on it.

```
mshgyedhmaedcrgdigrtnlivnylpqnmtqdelrslfssigevesaklirdkvaghslgygfvnyvtakdaeraintlglrlqsktikvsy  
arpssevikdanlyisglprtmtqkdvedmfsrfgriinsrvlvdqttglrgvafirfdkrseaeaaitsfnghkppgssepitvkfaanpnqnk  
nvallsqlyhsparrfggpvhgqaqrfspmgvdhmsglsgvnpqgnassgwcifiynlgqdadegilwqmfpgfavgatnkvirdfntn  
kckgfgfvmtnyeeamaiaaslngyrlgdkilqvsfktknkshk
```

Because in-gel digestion only involves trypsin, the company gave us suggestion to do combined digestion of trypsin, chymotrypsin and Glu-C, that will generate more possibilities to get the wanted segment of a readable peptide. Since the size of chymotrypsin and Glu-C is not suitable to do in-gel digestion, the company suggested us to prepare sample in soluble status. We are going to send out the sample to do mass spectrometry again following the company's suggestion. Yet, the company still claimed, with the modification size varying, the analysis might not reach our expectancy.

Thus, at this moment, we are not able to give the answer back to the reviewer. However, regarding the data to conclude the PARylation of HuR, we considered the analyses are sufficient and the modification is specific due to the application of both PARP1 inhibitor and PARG in the PARylation assays. We sincerely ask for reviewer overlooking this imperfection at the current stage.

It would be interesting to know if HuR functions also as a reader of PARylation, i.e., does HuR bind PAR in a non-covalent manner via specific PAR binding motifs?

Reply:

The reviewer's question also evoked our interest whether HuR may function as a PAR reader. We made effort to set up a protocol to address this issue.

First we need generate free PAR and valid the product. The *in vitro* PARylation system can generate autoPARylated PARP1, and then PARG involved, followed by immune depletion of PARP1. Western blotting verified, after PARG cleavage and immune depletion of PARP1, there is no PARP1 and autoPARylated PARP1 in the sample (note the second lane of the left panel in the below figure). Due to the small size of free PAR polymers, they run out of the gel after SDS-PAGE. Nevertheless, free PAR indeed exist in the sample because it can be detected out from the dot blotting assay (note the right panel in the below figure).

To detect whether HuR is a PAR reader, we did dot blotting. The dots of free PAR input serve as the control to verify the signal presenting the antibody recognition of PAR; GST was expected as a negative control; AIF, which was reported as a PAR reader (Yingfei Wang. et al. *Sci Signal*. 2011;4(167):ra20), serves as a positive control. We adjusted the

amount of recombinant proteins (see the lower part of the right panel) and titrated, and applied the same batches of recombinant proteins on the NC membrane twice for duplicates (upper part of the right panel). After 10% non-fat milk blocking, the membrane was incubated with free PAR for 1 h, then washed twice by TBST (each for 3 min). The bound PAR was cross-linked by UV irradiation (10 min), and then probed by antibody recognizing PAR.

The result showed AIF indeed is a PAR reader, but HuR not.

Regarding whether we display this result in the revised manuscript, we are considering, to avoid making the data set too complicated, should we not integrate it in manuscript.

In vitro free PAR binding assay. Purified GST, GST-HuR, GST-AIF proteins were incubated with Free PAR. Immuno-blotting was performed to detect the bound PAR levels. Recombinant GST and GST-fused proteins were stained with Coomassie brilliant blue. ☆, GST-AIF; *, AIF; #, GST.

The PARylation dependent shuttling mechanism should be analyzed in more detail, e.g., is it dependent on CRM1, as previously shown for p53?

Reply:

We agree with the reviewer's opinion that "The PARylation dependent shuttling mechanism should be analyzed in more detail". As we stated in DISCUSSION: "The involvement of PARylation in protein nuclear export has been addressed previously. RelA/p65 PARylation decreased its interaction with Chromosomal Maintenance 1 (CRM1, also known as Exportin 1) upon TLR4 stimulation, leading to NF- κ B nuclear retention, which ultimately influenced NF- κ B-dependent gene regulation⁴². Also, PARP1-mediated PARylation of p53 blocked the interaction of p53 with CRM1, resulting in the nuclear accumulation of p53⁴³. The export of HuR to the cytoplasm is regulated mainly in a CRM1-dependent manner through its association with nuclear ligands pp32 and APRIL, which contain nuclear export signals that are recognized by CRM1⁴⁴. Additionally, HuR

serves as an adapter for *c-fos* mRNA export through another pathway that involves the interaction of HNS with transportin 2⁴⁴. Thus, whether the downstream pathways mediating PARylated HuR shuttling from the nucleus to cytoplasm are CRM1-dependent or independent requires further investigation.

We considered that further exploring the mechanism how PARylated HuR undergoing nucleocytoplasmic shuttling, via CRM1-dependent or independent pathway, exceeds the scope of the present study. It indeed is what we are continuing to work on now.

Specific comments:

Fig. 1:

Panel a: Statistical significance could be tested via 2-way ANOVA analysis

Panel h: A PARP1 immunoblot should be included

Reply:

We followed the reviewer's suggestion, 1) tested the statistical significance via 2-way ANOVA analysis, which indicated PJ34 treatment significantly lowered LPS-induced increase in *Cxcl2* mRNA stability, with p value less than 0.001; 2) integrated the PARP1 immunoblot in the figure panel in the revised manuscript (see below).

Fig. 2:

Panel a: Molecular weight marker is missing

Panel b and e: A densitometric analysis of independent replicates should be included

Reply:

Following the reviewer's suggestions, we labeled the molecular weight marker with panel a; and did the densitometric analysis with panel b and e.

Furthermore, we also did the densitometric analysis with Fig. 3 a and b.

Fig.3:

Panel c: Why is automodified PARP1 protein not visible in the anti-PAR blot?

Reply:

Figure 3c showed automodified PARP1 protein.

Left panel: Bead-bound GST and GST-HuR were subjected to PARylation. After incubation, the beads were washed, so that the associated automodified PARP1 is less, shown as lane 3, the signals higher than 110 kDa are weak.

Right panel: Eluted GST and GST-HuR were subjected to PARylation. After incubation, the whole mixture containing automodified PARP1 was applied to immuno-blotting, thus, signals for automodified PARP1 are strong, shown as lane 8. Moreover, even incubation of GST with PARP1 also resulted in strong automodification of PARP1, shown as lane 6.

Fig.4: It is absolutely essential to quantify nuclear-cytoplasmic shuttling densitometrically via an imaging software (e.g., Image J), since the effect is hardly visible from the pictures shown. Quantitative data should be included from independent replicates.

Reply:

We thank reviewer for his/her pointing out the cytoplasmic localization of HuR is hardly visible from the pictures shown. We used ImageJ software to depict the individual signals in black-and-white to achieve the enhanced contrast. Since HuR is mostly located in nucleus in the normal cells, it's difficult to clearly define the cytoplasmic region, so we didn't show the quantitation by ImageJ analysis. We do agree it is absolutely essential to quantify nuclear-cytoplasmic shuttling, thus we did western blotting to quantify the cytoplasmic localization of HuR (Fig.4b and f).

Fig.4

Fig.6: The influence of free PAR polymer on HuR-RNA interaction should be analyzed

Reply:

Following reviewer's suggestion, we examined the influence of free PAR polymer on HuR-RNA interaction.

Free PAR was produced as described above. GST-HuR and PARylated GST-HuR were incubated with Cxcl2-ARE1 RNA oligo alone or in presence of titrated PAR. EMSA was conducted. Result showed free PAR polymers had impact on the interaction between PARylated HuR and RNA in a dose-dependent manner, but had no significant role in HuR/RNA interaction.

We are considering not integrating this data in the revised manuscript due to the following reasons:

- 1) The molecular mechanism by which PAR polymers impact the interaction between PARylated HuR and RNA could be an independent research project, needing many data to get conclusion. Integrating this data in manuscript will make the context more complicated;
- 2) Such impact is observed by introducing PAR polymers in an *in vitro* interaction. According to the *in vivo* scenario that intracellular PAR polymers are generated when DNA is highly damaged, and free PAR polymers are produced for acting as death messenger, the inflammation cell culture model in the present study may not generate an enormous amount of free PAR.

Taking the above reasons into account, we did not integrate the data in the revised manuscript currently.

But, what the reviewer suggested and what we found through the preliminary assay indicated a scientific issue that needs to further explore: How free PAR impact RNA-binding protein function, thereby gene expression in cells responding to DNA damage.

We will keep this in mind and make effort to further investigate in future.

Free PAR polymers have impact on the interaction between PARylated HuR and RNA in a dose-dependent manner. GST-HuR or PARylated GST-HuR was incubated with Cxcl2-ARE1 probe in presence of titrated free PAR or not. Gel retardation assay was performed.

Reviewer #2 (Remarks to the Author):

We appreciate the reviewer's recognition of our work. We also thank the reviewer for his/her criticism, which allows us improve the quality of the manuscript to meet the criteria of the journal.

While the experiments are well-designed and technically well-done, there are issues with the presentation of data. In particular, Figures 2 (RRM1, panel d; and panels e and f) and 6 (the whole figure) are messed up. The problems in Figure 6 do not allow for review of the data. The text in latter part of the Results section (from the sub-section "PARP1 enhances the binding of HuR to ARE-containing RNA" onward) is hard to follow, partly because of Figure 6 problems, but also because of syntax and wording.

Reply:

We re-labeled Figure 2 panel e and f to ensure the identity of the labeling within the figure, and easy to follow up the depiction.

Regarding the statement for Figure 6a, we deleted several words making the sentences shorter and easy to follow; and also add PARP1 in front of another sentence to make the sentence more precise. The related statement is as below:

The recombinant proteins were either subjected to PARylation or not, and then incubated with biotin-labeled tandem ARE repeat-containing RNA oligos **at different titration levels**. GST-HuR elicited several shifted bands, which might result from the different copies of GST-HuR harbored on the tandem ARE-containing probes (Fig. 6a, lanes 1–12), whereas GST failed to do so (Fig. 6a, lane 13). The incubation with PARP1 markedly enhanced the binding of GST-HuR with the probes (Fig. 6a, lanes 4–6) compared with GST-HuR ~~in~~ **PARylation buffer alone** (Fig. 6a, lanes 1–3), which was inhibited by PJ34 addition (Fig. 6a, lanes 7–9). The direct incubation of eluted GST-HuR with probes showed the same patterns of the shifted bands (Fig. 6a, lanes 10–12) as that of samples subjected to PARylation (Fig. 6a, lanes 1–9); in parallel, **as a vehicle control, PARP1** in PARylation buffer alone did not result in any shifted bands (Fig. 6a, lane 14).

We also made the related figure legend precise, thus easy to follow and matching with the statement in the text:

(a) Binding of the HuR with ARE motif-containing RNA oligo is enhanced by PARylation. *E. coli*-expressed GST-HuR or GST was purified and eluted, then subjected to PARylation or not. Differently conditioned proteins were titrated (20-, 10- and 5-fold diluted respectively) and then incubated with biotin-labeled tandem ARE repeat RNA oligos as indicated (upper panel). The gel retardation assay was performed to detect the binding of GST-HuR or GST to probes. The input amount of GST-HuR or GST within the binding system was visualized by Coomassie brilliant blue staining (lower panel).

Points that would strengthen the paper are:

1) Explain why PARG was used to inhibit PARP1 (at least provide a reference).

Reply:

We followed the suggestion of the reviewer, added the statement to explain why PARG was used to inhibit PARP1. The revised part of the relevant paragraph is as bellow (see the highlighted):

To further address which domain(s) and site(s) are PARylated, we developed an *in vitro* PARylation assay using GST-fused proteins as described in the Materials and Methods. First, bead-coated GST and GST-HuR were incubated with or without PARP1 in presence or absence of PJ34 or Poly(ADP-ribose) glycohydrolase (PARG), **the enzyme mediating the removal of ADP-ribose units from PARylated protein^{28,29}** (Fig. 3c, left).

Also the relevant referenced are shown as bellow:

28. Luo, X. & Kraus, W.L. On PAR with PARP: cellular stress signaling through poly(ADP-ribose) and PARP-1. *Genes Dev* 26, 417-432 (2012).

29. Mi Young Kim, Tong Zhang, and W. Lee Kraus. Poly(ADP-ribosyl)ation by PARP-1: 'PAR-laying' NAD⁺ into a nuclear signal. *Genes Dev.* 2005 19: 1951-1967

2) D226 is the cleavage site in HuR between CP1 and CP2. It is interesting that D226 is also the amino acid that, when mutated, disrupts HuR PARylation by PARP1. This coincidence and its possible implications should be discussed.

Reply:

We followed the suggestion of the reviewer, discussed the possible implications of D226 PARylation regarding the amino acid is also the site to be cleaved. The statement in the revised manuscript (highlighted) is as below:

An intriguing study demonstrated that upon lethal stress, HuR undergoes caspase-mediated cleavage at D226 in the cytoplasm. This cleavage activity is associated with the apoptosome activator pp32/PHAP-I, and this caspase-mediated cleavage constitutes a regulatory step that contributes to an amplified apoptotic response⁵¹. Here, we deduced that D226-mediated PARylation may impair the effect of the apoptosome and further influence HuR functions by slowing down its turnover rate in the cytoplasm.

3) Does HuR shuttle from nucleus to cytoplasm in the D226A mutant?

Reply:

HuR's shuttling from nuclear to cytoplasm is inhibited when aspartic acid 226 was mutated to alanine as shown in Fig.7c and d. To achieve the enhanced contrast, we used ImageJ software to depict the individual signals in black-and-white.

D226A HuR failed to shuttle to cytoplasm in response to LPS exposure

Other minor points:

1) Line 223 - states that there is a moderate increase in PARylated HuR in the NE fraction. It appears to be about the same increase as in CE

Reply:

We did the densitometric analysis with Fig. 3 a and b. The increase in PARylated HuR in the NE fraction is lower than that in CE. It may be due to the absence of HuR in CE in normal cells, and the consequence of nucleocytoplasmic shuttling of PARylated HuR.

2) Line 243 - There is no strong smear at the top of lane 3. Smear only present in lanes 6, 8, and 10.

Reply:

In left panel: bead-bound GST and GST-HuR were subjected to PARylation. After incubation, the beads were washed, so that the associated automodified PARP1 is less shown as lane 3; whereas, in right panel: eluted GST and GST-HuR were subjected to PARylation. After incubation, the whole mixture containing automodified PARP1 was applied to immuno-blotting, thus, signals for automodified PARP1 are strong, shown as lane 8. Moreover, even incubation of GST with PARP1 also resulted in strong automodification of PARP1, shown as lane 6. Less density of smear at the top of lane 10 was due to the application of PARG.

3) The figure legends tend to be excessively long and repeat information in the text.

Reply:

Following the reviewer's suggestion, we tried to make the figure legends more concise in the revised manuscript.

Reviewer #3 (Remarks to the Author):

We appreciate reviewer's positive comments and his/her recognition of our work. We also thank reviewer for the constructive suggestion that will improve the quality of the manuscript to meet the criteria of the journal.

Specific comments:

L. 152: There is literature on the destabilizing activity of Cxcl2 3'UTR, which the authors need to quote. E.g. Numahata et al. J Cell Biochem 2003 90:976.

Reply:

Following the reviewer's suggestion, we cited the paper in the revised manuscript.

21. Numahata, K. et al. Analysis of the mechanism regulating the stability of rat macrophage inflammatory protein-2 mRNA in RBL-2H3 cells. J Cell Biochem 90, 976-986 (2003).

L. 158: The authors do not show that the ARE mediates the PAPR1 effect, they can only conclude that the Cxcl2 3'UTR mediates the effect.

Reply:

We thank the reviewer for his/her advice that makes the depiction more accurate. We made this change in our revised manuscript (shown as below).

The combined data implied that **Cxcl2 3'UTR** mediate PARP1's regulation of mRNA stability.

L. 162: Ref. 21 is not appropriate for HuR as an ARE-mRNA stabilizing protein. The first reports were published by the Steiz and Shyu labs in 1998, both in EMBO Journal.

Reply:

We thank the reviewer for his/her suggestion on the appropriate citation. We integrated the related references in our revised manuscript.

22. Peng, S.S., Chen, C.Y., Xu, N. & Shyu, A.B. RNA stabilization by the AU-rich element binding protein, HuR, an ELAV protein. EMBO J 17, 3461-3470 (1998).

24. Fan, X.C. & Steitz, J.A. Overexpression of HuR, a nuclear-cytoplasmic shuttling protein, increases the in vivo stability of ARE-containing mRNAs. EMBO J 17, 3448-3460 (1998).

The authors frequently use descriptive wording when reporting on the effects observed. E.g. "severely impaired" (L.149), "significantly in/decreased" (L.151, 155), "markedly diminished" (L.157), and many more examples throughout the text. Instead, the authors should provide exact numbers, e.g. 20-fold increase, 3.4-fold decrease etc.

Reply:

Following reviewer's suggestion, we quantified the alterations stated in L.149, 151, 155, 157. We also extended this quantification to Fig. 1h.

Fig. 1f shows only a relatively small effect of LPS on Cxcl2 3'UTR reporter mRNA levels (less than two-fold). The authors should confirm by Act.D chase experiments (as in Fig.1a,b) that the Cxcl2 3'UTR reporter mRNA is indeed stabilized by LPS treatment. Since this is a qPCR after transient transfection of the reporter plasmid, the authors should document that the signal they measure is derived from RNA, and not a plasmid DNA contamination in the RNA preparation. This can be done by a minus-RT control.

Reply:

The reporter gene is driven by SV40 promoter (pGL3-control plasmid-based), thus there is no influence variation among samples at transcriptional level. Yet, there might be plasmid contamination for qPCR as the reviewer concerned. Following reviewer's suggestion, we re-did experiments for Fig.1f and h with utilization of DNase to remove the impact of plasmid DNA. The results showed the mRNA levels of reporter gene tagged with Cxcl2-3'UTR increased ~5 in response to LPS exposure (Fig.1f). (shown as below):

Regarding utilization of DNase, we also modified the related statement in Materials and Methods.

To further analyze the effects of the 3'UTR on mRNA levels of target genes (RNA was extracted (1U DNase is used in 1µg RNA to eliminate plasmid DNA contamination)). Firefly luciferase mRNA levels were measured by real time PCR and calibrated to that of Renilla.

Fig. 1g: The data in panels 1e+f suggest that PARP1-dependent mRNA stabilization is responsible for a less than 2-fold increase in Cxcl2 reporter mRNA levels. In contrast, kd of

HuR diminished Cxcl2 mRNAs levels about 6-fold in panel 1g. Does this mean that HuR also has an effect on transcription of Cxcl2?

Reply:

As we state above, we re-did experiments for Fig.1f and the results showed the mRNA level of reporter gene tagged with Cxcl2-3'UTR increased ~5 folds in response to LPS exposure. Thus data of Fig.1f and 1g are compatible.

HuR has been implicated in multiple other RNA metabolism processes, such as alternative pre-mRNA splicing and protecting mature mRNA for nuclear export; however, so far, there is no evidence indicating the role of HuR in gene transcription.

L. 186: I do not understand why "The involvement of whole cell lysate input and molecular weight standard" should indicate "that the interaction of two molecules relies on their 188 full-length forms".

Reply:

The reviewer's concern prompted that our statement is somewhat misleading. Both HuR and PARP1 are targets of caspases, the cleavage of the full-length 116 kD PARP1 produces the 85-kD, and 36 kD HuR to 24 kD forms. We used the whole cell lysate and molecular marker to indicate that the interaction of two molecules relies on full-length PARP1 (116 kD) and HuR (36 kD). We re-wrote the related lines shown as below:

Both PARP1 and HuR are targets of caspases, and may undergo cleaved **under stress**. **Thus the whole cell lysate input was applied and molecular weight standard was laid out in the western blot, the result** indicated that the interaction of two molecules relies on their full-length forms (Fig. 2b and c).

L. 200: The term "abundantly pulled down" is not warranted, since binding of HNS and RRM3 is clearly below binding of WT HuR (Fig. 2e).

Reply:

We modified the related depiction as below:

GST-HuR-HNS- and GST-HuR-RRM3-fused proteins **could modestly** pull down PARP1; whereas GST-HuR-RRM1 and GST-HuR-RRM2 hardly showed **such** interaction (Fig. 2e).

Fig. 3a: The amount of HuR in the PAR-IP (right panel) is barely above the background level. The authors should either omit this experiment, or show a thorough quantification.

Reply:

In this figure, due to the overload of input, we had to choose a very light exposure to avoid

over-saturated signal of input HuR, resulting in that “The amount of HuR in the PAR-IP (right panel) is barely above the background level.” So we deleted the input lane here, and increased the exposure. We also quantified the band intensities. The result showed the amount of HuR in the PAR-IP increased **1.55** folds in response to LPS exposure.

Fig. 4: In many of the IFs, the cytoplasmic localization of HuR upon LPS treatment is difficult to see. For one thing, the authors should depict the individual signals in black-and-white, which enhances the contrast, and use colors only in the merged panels. Second, it would be helpful to show larger magnifications, and increase the size of the image depicted. Third, and most importantly, the authors need to quantify the proportion of cytoplasmic HuR in the IFs by image analysis, and provide numbers in addition to the merely descriptive images. This comment also applies to Fig. 7c and d.

Reply:

Following reviewer’s suggestion, we used ImageJ software to depict the individual signals in black-and-white to achieve the enhanced contrast. Since HuR is mostly located in nucleus in the normal cells, it’s difficult to clearly define the cytoplasmic region, so we didn’t show the quantitation by ImageJ analysis. We do agree it is absolutely essential to quantify nuclear-cytoplasmic shuttling, thus we did western blotting to quantify the cytoplasmic localization of HuR (Fig.4b and f). Regarding Fig.7c and d, as the magnificence is larger than that of Fig.4b and f, signals in black-and-white can clearly show the cytoplasmic location of WT HuR and K191A HuR in response to LPS exposure, which was not shown with D226A HuR.

Fig.4

Fig.7

Fig. S5: I am not convinced by this experiment since the amount of overexpressed Flag-HuR is way below the level of endogenous HuR, and Flag-HuR cannot even be seen in the PARP1 IP. I suggest to omit this experiment.

Reply:

Following reviewer's suggestion, we omitted this experiment and related statement in our revised manuscript.

Fig. 5: From these RNA-IPs, the authors cannot conclude that the interaction of HuR with Cxcl2 mRNA changes in an LPS- and PARP-dependent manner, because the Cxcl2 mRNA levels (input) change in a similar manner, as shown in Fig. 1g. To assess binding activity, the authors need to normalize the RNA signal in the HuR-IP to the amount of Cxcl2 mRNA in the input, and, to be fully accurate, also to the amount of HuR in the IP. This comment also

applies to Fig. 7b.

Reply:

The concern of the reviewer is quite rational. Indeed, as shown in Fig. 1g, level Cxcl2 mRNA was resulted from both transcriptional and posttranscriptional input. We took this scenario into the consideration, thus in the present study, to do RNA-IP, the cells were exposed to 500 ng/mL LPS for 1 h to boost pro-inflammatory gene expression followed by addition of Act D. Meanwhile the cells were withdrawn from LPS, maintained in LPS during incubation, or treated with LPS plus PJ34 for another 2 h. Thereafter, the RNA-IP was performed.

The reviewer’s criticism reminded us, we missed the important information on the addition of Act D in figure legends and method.

Also, legend for Fig. 7b is not appropriate. We revised the related depiction, and also, we added mRNA input in the panels.

Fig.5

Fig. 6a: While the data indicate enhanced binding of ARE-RNA to PARylated HuR compared to unmodified HuR, the authors should take their EMSAs a step further and calculate Kd values for both forms of HuR. For this, they will need to increase the concentration of HuR until they reach saturated binding. Moreover, it would be interesting to know the Kd of the D226A mutant (in presence of PARP1).

Reply:

Following reviewer’s suggestion, we performed EMSA to compare the binding of Wt and D226A HuR to Cxcl2-ARE1 RNA oligo. We further tried to calculate the Kd values according to the method described in the previous publication (von Roretz, C. *et al.* Apoptotic-induced cleavage shifts HuR from being a promoter of survival to an activator of caspase-mediated apoptosis. *Cell Death Differ* 20, 154-168 (2013).

The result showed, after incubation with PARP1, binding of Wt HuR to Cxcl2-ARE1 RNA oligo was accelerated to more than 2 folds (Kd value decreased more than 50%), while D226A mutant displayed the affinity kinetics similar as that of Wt HuR in absence of PARP1.

We integrated this data in RESULT section, the below figure was presented as Fig. 6h

h. Incubation with PARP1 enhances the interaction of Wt HuR but not D226A HuR with Cxcl2 ARE1-containing RNA. Varying concentrations of GST-HuR or GST-HuR-D226A was incubated with PARP1 or not, Gel-shift was performed as described above. Quantification of bound and unbound signal allowed dissociation constants (Kd) to be determined³¹.

Fig. 6f: According to the scheme in 6e, ARE1 and ARE3 are identical. Why is binding of the two oligos to HuR so different in 6f? The authors should depict the actual oligo sequences that were used in the EMSAs.

Reply:

As the reviewer mentioned, the sequence of ARE1 and ARE3 are identical but the affinity for HuR binding is quite different. Not only that, our expectancy is ARE2 might trap more HuR but the result showed more retardant complex formed with ARE1. Obviously, like the reviewer questioned, the surrounding context of ARE is supposed to have impact that remains uncovered.

Here, we followed the reviewer's suggestion, putting the exact sequence of each probe in the Fig.6 e shown as below:

L. 372: The authors can only talk about Cxcl2 mRNA, not about mRNAs from pro-inflammatory genes in general, since they only measured Cxcl2 mRNA in Fig. 7f.

Reply:

We changed the statement as the reviewer suggested.

Reviewers' Comments:

Reviewer #1 (Remarks to the Author)

All my points have been satisfactorily addressed by the authors. I recommend publication of the manuscript as it is.

Reviewer #2 (Remarks to the Author)

The authors have adequately addressed the comments and concerns of the previous review. I agree with those cases where the authors have opted not to include newly generated data requested by the reviewers so as to avoid defusing the main focus of the work.

Reviewer #3 (Remarks to the Author)

My two major concern have not been addressed by the authors since proper quantification of HuR localization and RNA-binding are still lacking. Therefore, I am not fully convinced by the authors conclusions, and I cannot recommend publication of the manuscript.

One of my major concerns was that the PARP-dependent export of HuR into the cytoplasm needs to be quantified properly (Fig.4 and 7). However, no efforts were made to quantify the nuclear / cytoplasmic distribution of HuR. Without an objective quantification of the micrographs and the western blots of fractionation experiments, the changes, e.g. upon PARP1 knockdown (Fig. 4e), are not convincing.

A second major concern was the claim that binding of HuR to Cxcl2 mRNA is enhanced by PARylation. The data depicted in Fig. 5 do not warrant such a statement because the amount of Cxcl2 mRNA in the HuR IP parallels that in the input (shown in Fig. 1g). From their response, the authors seem to be unwilling to conduct proper normalization by dividing the amount of RNA in the IP by the amount in the input.

Similarly, the EMSA results shown in Fig. 6 are technically not optimal, and I strongly doubt that reliable Kd values can be derived from Fig. 6h. For proper EMSAs, one has to increase the protein amount such that saturated binding is achieved, and one has to plot the fraction of bound RNA over the different protein concentrations. In fact, saturation might have been achieved for GST-HuR without PARP (first four lanes of Fig. 6h), whereas saturation is clearly not achieved in the other two reactions (lanes 5-8 and 9-12). If anything, this would suggest that the Kd of GST-HuR without PARP is lower than the Kd of GST-HuR with PARP. Indeed, at the second concentration, one can see more binding of HuR in the absence of PARP (lane 2) than in the presence of PARP (lane 6). This would suggest that HuR binds RNA more tightly in the absence of PARP, which is the opposite of what the authors claim.

English language remains an issue in the revised manuscript.

Reviewer #3 (Remarks to the Author):

My two major concern have not been addressed by the authors since proper quantification of HuR localization and RNA-binding are still lacking. Therefore, I am not fully convinced by the authors conclusions, and I cannot recommend publication of the manuscript.

Reply:

We are grateful for the reviewer's constructive suggestions, which help us substantially improve the quality of the manuscript for the possible publication by ***Nature Communications***. The two major concerns of the reviewer---- (1) proper quantification of HuR localization; (2) Kd values of RNA-binding-----have been answered (see below). We hope that the reviewer would be satisfied with our revised manuscript.

One of my major concerns was that the PARP-dependent export of HuR into the cytoplasm needs to be quantified properly (Fig.4 and 7). However, no efforts were made to quantify the nuclear / cytoplasmic distribution of HuR. Without an objective quantification of the micrographs and the western blots of fractionation experiments, the changes, e.g. upon PARP1 knockdown (Fig. 4e), are not convincing.

Reply:

Following the reviewer's suggestion, we quantify the nuclear / cytoplasmic distribution of HuR by densitometry using Image J software (version 1.44). We measured the total HuR amount first, and then that of nuclear HuR, thus we had the cytoplasmic amount of HuR by taking nuclear amount of HuR away from that of the total. The re-distribution of HuR was estimated by dividing the cytoplasmic amount of HuR by that of total.

The implicated original figures are Fig. 4 a, c, d and e, as well as Fig. 7 c and d. The revised figures are shown as below:

Fig.4

Fig.7

Also we followed the reviewer's suggestions, supplemented western blotting data to support the immuno-fluorescence results. The related figures are integrated in supplementary materials, presented as Fig. S4 d,f and Fig. S6 (See below).

Fig. S4

Fig.S6

Per the addition of graphical depiction of quantification on the nuclear / cytoplasmic distribution of HuR, we re-arranged Fig. 4, moving original Fig. 4b and f to supplementary materials (presented as Fig. S4b and g).

A second major concern was the claim that binding of HuR to Cxcl2 mRNA is enhanced by PARylation. The data depicted in Fig. 5 do not warrant such a statement because the amount of Cxcl2 mRNA in the HuR IP parallels that in the input (shown in Fig. 1g). From their response, the authors seem to be unwilling to conduct proper normalization by dividing the amount of RNA in the IP by the amount in the input.

Reply:

The reviewer's concern is very rational because of the consideration of the amount of Cxcl2 mRNA in the inputs

We realized that we did not depict clearly, that misled the reviewer.

When we conducted RNA-immunoprecipitation (RNA-IP) assays, we had cells exposed to 500 ng/mL LPS for 1 h to boost pro-inflammatory gene expression and then Act D was added with or without LPS maintenance (\pm PJ34) for another 2 h. ("addition of Act D" is missing, and we added this information in our revised manuscript). Thus, the input amount of Cxcl2 mRNA was not parallel as shown in Fig. 1g. Because of conducting assays as described above, the amount of Cxcl2 mRNA in the inputs did not vary greatly (see below).

Moreover, the way we calculated the amount of Cxcl2 mRNA in the HuR IP is same as what the reviewer suggested: “normalization by dividing the amount of RNA in the IP by the amount in the input”. (We integrated this statement in our revised manuscript).

The related statement in the text is shown as below:

“Thus, we performed RNA-immunoprecipitation (RNA-IP) assays to investigate the role of PARylation in the binding of HuR to target mRNA. Cells were exposed to 500 ng/mL LPS for 1 h to boost pro-inflammatory gene expression and then Act D was added with or without LPS maintenance (\pm PJ34) for another 2 h. HuR-associated RNA target was normalized by dividing the amount of RNA in the IP by that in the input. From the whole cell lysate of mock cells, the binding of HuR to *Cxcl2* mRNA was barely detectable; however, from the LPS-challenged cell lysate, abundant *Cxcl2* mRNA was pulled down. The interaction of HuR with *Cxcl2* mRNA was inhibited by PJ34 (Fig. 5a).”

We realized the reason we misleading the reviewer may be also due to the inappropriate labeling of the y axis of the histogram. We change the title of y axis as “Cxcl2 mRNA level (IP-ed/Input)” instead of “Cxcl2 mRNA level in IP”. We might also not label the samples appropriately. Thus, we changed the sample labeling shown as below.

Fig.5

Similarly, the EMSA results shown in Fig. 6 are technically not optimal, and I strongly doubt that reliable K_d values can be derived from Fig. 6h. For proper EMSAs, one has to increase the protein amount such that saturated binding is achieved, and one has to plot the fraction of bound RNA over the different protein concentrations. In fact, saturation might have been achieved for GST-HuR without PARP (first four lanes of Fig. 6h), whereas saturation is clearly not achieved in the other two reactions (lanes 5-8 and 9-12). If anything, this would suggest that the K_d of GST-HuR without PARP is lower than the K_d of GST-HuR with PARP. Indeed, at the second concentration, one can see more binding of HuR in the absence of PARP (lane 2) than in the presence of PARP (lane 6). This would suggest that HuR binds RNA more tightly in the absence of PARP, which is the opposite of what the authors claim.

Reply:

We are very grateful for the reviewer giving us the elaborate instruction on how to perform the proper EMSAs to estimate K_d values, which led us fully understand the significance of K_d value calculation.

Following the reviewer's suggestion, we re-did the EMSA and estimates the K_d values of different samples. The result was presented as Fig. 6h, and shown as below.

Fig. 6

English language remains an issue in the revised manuscript.

Reply:

Following the reviewer's suggestion, we purchased editing service from a company (International Science Editing, Shannon, Co Clare, Ireland). The version with the editing track, which the reviewer may refer to, was submitted along with a clean version.

Reviewers' Comments:

Reviewer #3 (Remarks to the Author)

The authors have substantially revised their manuscript and addressed my essential concerns.